# GENERALIZABLE COARSE-TO-FINE ROBOT MANIPULATION VIA LANGUAGE-ALIGNED 3D KEYPOINTS

**Jianshu Hu, Lidi Wang, Shujia Li, Yunpeng Jiang, & Yutong Ban**[†]
Global College
Shanghai Jiao Tong University
{hjs1998, lidiwang, sjtu7264970, jyp9961, yban}@sjtu.edu.cn

**Xiao Li**
School of Mechanical Engineering
Shanghai Jiao Tong University
{sjtu_lixiao}@sjtu.edu.cn

**Paul Weng**[†]
Digital Innovation Research Center
Duke Kunshan University
{paul.weng}@dukekunshan.edu.cn

## ABSTRACT

Hierarchical coarse-to-fine policy, where a coarse branch predicts a region of interest to guide a fine-grained action predictor, has demonstrated significant potential in robotic 3D manipulation tasks by especially enhancing sample efficiency and enabling more precise manipulation. However, even augmented with pre-trained models, these hierarchical policies still suffer from generalization issues. To enhance generalization to novel instructions and environment variations, we propose Coarse-to-fine Language-Aligned manipulation Policy (CLAP), a framework that integrates three key components: 1) task decomposition, 2) VLM fine-tuning for 3D keypoint prediction, and 3) 3D-aware representation. Through comprehensive experiments in simulation and on a real robot, we demonstrate its superior generalization capability. Specifically, on GemBench, a benchmark designed for evaluating generalization, our approach achieves a 12% higher average success rate than the SOTA method while using only 1/5 of the training trajectories. In real-world experiments, our policy, trained on only 10 demonstrations, successfully generalizes to novel instructions and environments.

## 1 INTRODUCTION

Robot learning, especially via imitation learning, has demonstrated promising success in enabling robots to solve complex 3D manipulation tasks (Intelligence et al., 2025; Liu et al., 2024). However, scaling these methods to a broader range of real-world applications (e.g., industrial, service, or home robotics) requires enhancing both (G1) their generalization to environment variations, and (G2) their skill compositional generalization. Indeed, G1 is necessary, because deployed robots need to be able to operate in new settings (e.g., object or background variation), while G2 is highly desirable, so that trained robots can tackled new tasks by composing previously-learned skills. To achieve G1 and G2, the robot needs to be endowed with a combination of capabilities, such as scene understanding, reasoning or planning, and high-precision manipulation, exploiting preferably sample efficient techniques, since robotics data is costly to collect.

In this paper, we focus on one type of 3D manipulation policies, called *coarse-to-fine* policies (Gualtieri & Platt, 2020; James et al., 2022; Ling et al., 2024; Goyal et al., 2024; Gervet et al., 2023; Liu et al., 2025), because they achieve superior precision in manipulation tasks while enjoying strong sample efficiency. These policies process 3D observations (or 3D scene representations) using a hierarchical architecture whose higher-level coarse branch identifies a region of interest for the lower-level fine-grained branch to focus on and predict a final action. Typically, the coarse branch is trained to predict a 3D keypoint, which serves as the center for cropping and zooming into the original 3D observations. To help with visual understanding and to some extent spatial

---

[†]indicates corresponding author

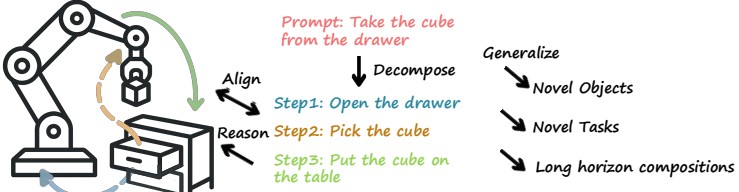

Figure 1: **Intuition of CLAP.** Our method achieves strong generalization ability by decomposing tasks into step-wise language instructions, each aligned with a 3D keypoint.

reasoning, recent work (Li et al., 2025b; Fang et al., 2025) has extended this approach to exploit pre-trained models—Vision-Language Models (VLMs) (Beyer et al., 2024) or visual foundation models (Ravi et al., 2025). However, the performance of these obtained methods is still limited in terms of generalization capability (G1 and especially G2), indicating that their scene understanding and reasoning capabilities are actually still rudimentary. Our experimental study suggests that this is primarily due to a combination of various issues (depending on the method), such as domain shift between pre-training and robotic images, inadequacy of pre-trained models to predict 3D keypoint, poor adaptation to object variations, or under-exploitation of the planning ability of VLMs.

To address these limitations and issues, we propose Coarse-to-fine Language-Aligned manipulation Policy (**CLAP**), a novel coarse-to-fine 3D manipulation policy. In contrast to previous coarse-to-fine policies, CLAP includes a novel architecture for the higher-level branch, which we name *coarse task planner*, and a novel implementation of the lower-level fine-grained action predictor, both leveraging pre-trained models.

The coarse task planner, implemented as a VLM, is introduced to play the additional role of task planning. Before the usual 3D keypoint prediction, it decomposes a task into step-wise language instructions, representing basic skills. This change allows both 3D keypoint and action predictions to depend on step-wise instructions instead of the whole task description, which promotes skill compositional generalization (G2). The training of this coarse task planner consists of three parts to reinforce its scene understanding and reasoning capabilities. First, the pre-trained VLM fine-tuned on language plans of different tasks to directly improve compositional reasoning. Second, it is specialized for 3D keypoint prediction by fine-tuning it to perform a sequential reasoning process: first localizing task-related objects, then generating the step instruction, and finally predicting a corresponding 3D keypoint. Finally, to further boost its scene understanding capability, the VLM is further fine-tuned with an auxiliary task of 3D object detection, using an additional dataset of object positions. Together, these components form a comprehensive pipeline that significantly enhances the generalization ability of the coarse-to-fine policy to object variations (G1) and novel tasks (G2).

The fine-grained action predictor takes as input both the step instruction and the multi-view RGB-D images and outputs an action. It is implemented with specialized pre-trained models to improve sample efficiency and increase its precision during manipulation. More specifically, step instruction and RGB images are processed using a pre-trained visual-language encoder, ensuring the two modalities are well-aligned. The depth information is processed by a dedicated encoder and augmented with 3D position embeddings to help better align 3D and 2D image information. All the obtained embeddings, which we call 3D-aware representation, are fused via a Multi-View Transformer (Goyal et al., 2023) to predict the final actions.

To evaluate the performance of our method, we run experiments in both simulation and real-world. For simulation, we use GemBench (Garcia et al., 2025), a benchmark specifically designed to assess the generalization ability of multi-task language-conditioned policies across varying difficulty levels. Our approach outperforms the state-of-the-art method, achieving a 12% higher average success rate with only 1/5 of the training trajectories. In real-world experiments, our method demonstrate strong generalization ability to novel tasks and object variations with only 10 demonstrations per task.

**Contributions**

1. We introduce a novel coarse-to-fine 3D manipulation policy, as shown in Figure 1, with two main innovations: (1) tasks are decomposed into step-wise language instructions to

promote compositional generalization ability; (2) action inference is performed via a reasoning step to improve generalization to object variations.

2. We design a finetuning pipeline that effectively adapts a pre-trained VLM to 3D keypoint prediction and incorporate a 3D-aware representation in the fine-grained action predictor, overcoming the issues observed in previous methods.

3. Empirical evaluations in simulation and on a real robot demonstrate state-of-the-art performance in both robustness to visual and object changes and generalization to unseen tasks.

## 2 RELATED WORK

In this section, we discuss the related works in the field, including vision-language-action models, 3D manipulation policies, and coarse-to-fine policies.

**Vision-Language-Action (VLA) models**  Training VLMs (OpenAI, 2024; Beyer et al., 2024; Bai et al., 2025; AI, 2024) on vast internet-scale image-text corpora has led to remarkable capabilities in image understanding, excelling at tasks like image classification, object detection, and visual question answering tasks. However, applying a similar training strategy directly to robotics presents a challenge due to the relatively scarce robot trajectory data. A prominent solution is to transfer the knowledge from pre-trained VLMs by fine-tuning them on robot data. This approach is the foundation for recent VLA models (Driess et al., 2023; Brohan et al., 2023; Kim et al., 2024; Octo Model Team et al., 2024; Intelligence et al., 2025; Wen et al., 2025a; Shukor et al., 2025; Liu et al., 2024; Wen et al., 2025b; Li et al., 2024; Cheang et al., 2025; Team et al., 2025; NVIDIA et al., 2025), which are fine-tuned on large diverse datasets of robot trajectories. Such extensive training strengthens generalization to novel objects, environments, and tasks. However, since they commonly use multi-view 2D images as visual input, learning to reason in 3D space from 2D images alone is data-intensive. This leads to sample inefficiency and low success rates on some tasks. Recent work has sought to more explicitly incorporate 3D information (Li et al., 2025a; Qu et al., 2025; Zhen et al., 2024) or introduce Chain of Thought (Mu et al., 2023; Zawalski et al., 2024; Zhao et al., 2025) to enhance the 3D reasoning ability. However these directions remain relatively underexplored within the VLA paradigm. Our method, which fine-tunes a pretrained VLM as a coarse task planner and predicts the final action with a fine-grained action predictor, can also be viewed as a VLA model. In contrast to other VLA approaches, we propose specific training and inference techniques to better align pre-trained VLMs to 3D manipulation, further enhancing generalization (G1-G2) while retaining the sample efficiency inherent to hierarchical coarse-to-fine policies.

**3D Manipulation Policy**  3D manipulation policies (Shridhar et al., 2022; Gervet et al., 2023; Jia et al., 2025; Zhu et al., 2025; Wang et al., 2024b; Ze et al., 2024; Wang et al., 2024a; Goyal et al., 2024; Ke et al., 2024; Chen et al., 2025a;b; Fang et al., 2025; Li et al., 2025b; Garcia et al., 2025) directly work with 3D inputs and outputs. They generally include structured architectures that construct a 3D representation of the scene, leading to higher sample efficiency and better generalization to new camera viewpoints. For example, PerAct (Shridhar et al., 2022) explicitly represents the scene with a voxel representation. Gervet et al. (2023) and Ke et al. (2024) process RGB images with pre-trained image encoder and lift 2D features to 3D by aggregating with depth information. An alternative approach (Jia et al., 2025; Goyal et al., 2024) is to project point clouds into canonical virtual views and use the resulting multi-view images as input. Explicitly exploiting 3D information allows these models to achieve high success rates with much less training data, which can be further reduced by enforcing a hierarchical structure like in coarse-to-fine policies.

**Coarse-to-fine Policies**  Gualtieri & Platt (2018; 2020) first propose this coarse-to-fine scheme for pick-and-place tasks. Subsequent work has considered more general tasks and explored various 3D representations, such as voxel observations (James et al., 2022; Liu et al., 2025), 3D feature fields (Gervet et al., 2023), and multi-view images (Goyal et al., 2024). Ling et al. (2024) apply the coarse-to-fine architecture to handle noisy point clouds. Among these, Robotic View Transformer 2 (RVT2) (Goyal et al., 2024) is an effective language-conditioned multi-task policy, demonstrating strong performance in both training and inference efficiency by using multi-view images projected from canonical views. However, RVT2 is trained from scratch, which limits its generalization ability to visual perturbations and task variations. Subsequent efforts built upon this work have sought to

overcome these limitations. Existing works (Li et al., 2025b; Fang et al., 2025) have attempted to enhance generalization through strategies such as: pre-training on object detection datasets (Yuan et al., 2024) or integrating encoders from powerful visual foundation models like Segment Anything Model 2 (Ravi et al., 2025). In contrast, we achieve this by introducing a novel architecture, where tasks are decomposed into step-wise language instructions for skill compositional generalization and design a specific training and inference pipeline to leverage pre-trained models in both coarse and fine-grained branches.

## 3 BACKGROUND

In this section, we first briefly recall the multi-task imitation learning set-up, introduce coarse-to-fine policy, and then present Robotic-View-Transformer 2 (RVT2) (Goyal et al., 2024), a state-of-the-art coarse-to-fine policy that serves as the foundation for our method.

In multi-task imitation learning, a dataset $\mathcal{D} = \{(\tau_i, L_i) \mid i = 1, ..., N\}$ is available for pairs of robot demonstrations $\tau_i$ and task description $L_i$. A robot demonstration is a trajectory $\tau_i = (o_0, a_0, o_1, a_1, ...)$ containing a sequence of observations $o_t$ and corresponding expert actions $a_t$. Observations include multi-view RGB-D images and gripper status indicating whether it is close or open. Actions denote the state of the end-effector, which contains the 3D position $p_t = (x_t, y_t, z_t)$ of the gripper, the orientation of the gripper and a gripper status.

A coarse-to-fine policy contains a coarse branch and a fine-grained branch, where the coarse branch predicts a 3D keypoint as the center to zoom in the 3D observation and the fine-grained branch uses the refined observation to predict the target action. Such policy is trained according to the key-frame based imitation learning framework (Johns, 2021; Shridhar et al., 2022; Goyal et al., 2024). Specifically, key-frames identifies timesteps in a trajectory when an important action, like grasping or placing, occurs. In practice, they are usually heuristically defined for each trajectory. With these key-frames, a trajectory is segmented into $K$ subsequences of observations and actions $(o_0, a_0, ..., o_{t_1}, a_{t_1}), ..., (o_{t_{K-1}+1}, a_{t_{K-1}+1}, ..., o_{t_K}, a_{t_K})$, where the $k^{th}$ key-frame occurs at time step $t_k$, from which we can extract a sequence of key-frame actions $(a_{t_1}, ..., a_{t_K})$. In this framework, the goal is to train a policy $\pi$ to predict the key-frame action $a_{t_k}$ at the closest next key-frame of timestep $t_k$ given an observation $o_t$ and a task description $L_i$:

$$\pi(o_t, L_i) \rightarrow a_{t_k} \text{ for } t_{k-1} \leq t < t_k. \tag{1}$$

The predicted actions are executed by a motion planner, which moves the robot to the desired state, generating thus the intermediate actions in a trajectories. In coarse-to-fine policies, the 3D position $p_{t_k}$ output by the coarse branch for the next key-frame is typically used as the 3D keypoint to zoom into the observation for the action predictor.

In RVT2, multi-view RGB-D images are first aggregated into a point cloud, which is then projected into three canonical views: front, left and top. These three views are orthogonal to each other, which allows a mapping between pixel positions in these views and a 3D position in the scene. Each pixel in the projected images contains 3-channel RGB values, 1-channel depth value and its corresponding 3D position in the global coordinate. In the coarse branch, the projected images are tokenized using convolutional layers while the task description and robot states (e.g., gripper status) are encoded by a pre-trained language encoder and a trainable Multi-Layer-Perceptron respectively. All these tokenized features are fused via Multi-View Transformer (Goyal et al., 2023). The image tokens in the output of the transformer are then processed by upsampling layers to predict heatmaps, from which a 3D keypoint is extracted. The keypoint from the coarse branch is used to zoom in and crop the point cloud while the cropped region is again projected into the canonical views. The refined observations along with the same task description are processed by the fine-grained branch, implemented as another multi-view transformer with different weights, to predict the final actions. While RVT2 achieves strong sample efficiency and enables precise manipulation via projections to canonical views and its coarse-to-fine architecture, it is trained from scratch and therefore does not leverage recent pretrained large models. In addition, its architecture design does not fully exploit common skills among tasks. As a result, it suffers from deficient generalization to visual changes, object variations, and novel tasks.

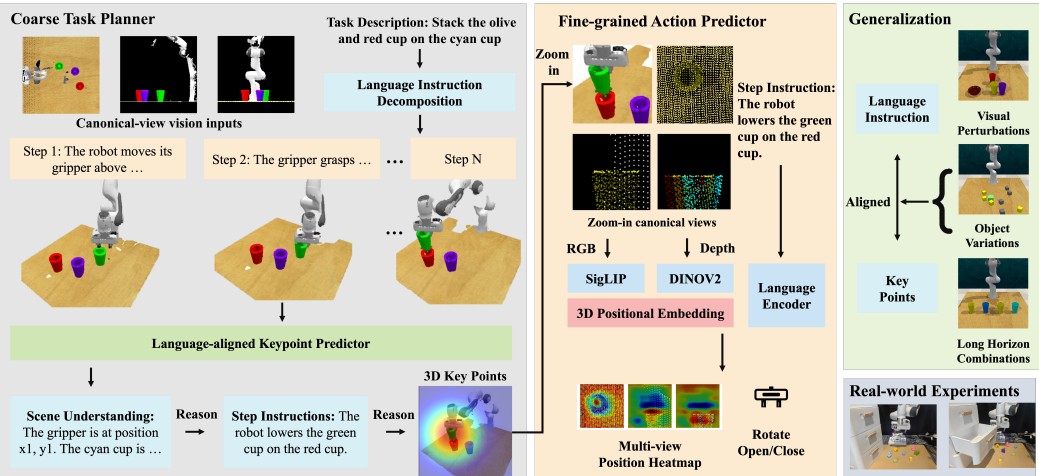

Figure 2: **Overview of CLAP.** We propose a novel coarse-to-fine 3D manipulation policy, comprising of a coarse task planner and a fine-grained action predictor. The coarse task planner reasons about the task plans and the positions of task-related objects to generate language-aligned 3D keypoints. The fine-grained action predictor fuses the corresponding step instruction with a 3D-aware visual representation from refined observations to predict the final action.

# 4 METHOD

Our hierarchical policy consists of a coarse task planner and a fine-grained action predictor, as shown in Figure 2. To promote compositional generalization, we first decompose tasks into step-wise language instructions, each describing the motion of the robot between two consecutive key-frames. This enables the coarse task planner to perform language-guided planning while allowing the fine-grained action predictor to learn and reuse common skills across tasks. Moreover, we adapt a pre-trained VLM to 3D keypoint prediction in the coarse task planner, by finetuning it with a sequential reasoning procedure. The VLM predicts the pixel position in each view and we follow the prior work (Goyal et al., 2024) to map them to a 3D position. The pretrained VLM is trained to first reason about the positions of task-related objects, then generate a step instruction and finally predict the corresponding 3D keypoints. To enable zero-shot generalization ability to novel objects, we add an auxiliary task of 3D object detection by augmenting the training data with additional object positions dataset. Finally, in contrast to RVT2, the coarse and fined-grained branches are implemented with a different architecture, since they play different roles. Thus, for the latter, we utilize specialized pre-trained visual foundation models to construct a 3D-aware representation. The design choices of both the coarse task planner and the fine-grained action predictor are detailed in the following sections.

## 4.1 COARSE TASK PLANNER

Prior coarse-to-fine policies condition all actions within a trajectory on a single high-level task description, limiting compositional generalization. To address this, we leverage a pre-trained VLM, denoted $f_\theta$, to decompose a high-level task description $L$ into step-wise language instructions $\mathcal{L} = (\ell_1, ..., \ell_k, ..., \ell_K)$, which naturally align with the key-frame based trajectory segmentation. Each trajectory segment $(o_{t_{k-1}+1}, a_{t_{k-1}+1}, ..., o_{t_k}, a_{t_k})$ and key-frame action $a_{t_k}$ have a corresponding step instruction $\ell_k$ describing the motion of the robot within the segment. For example, the task "open the top drawer" is decomposed into following step instructions: $\ell_1$: "The robot arm lowers itself to align with the handle of the top drawer", $\ell_2$: "The robot arm grasps the top drawer's handle firmly", and $\ell_3$: "The robot pulls the handle back, smoothly opening the top drawer".

In addition to task decomposition done at the beginning, at every execution timestep, the VLM $f_\theta$ is also exploited to predict both the step instruction $\ell_{t_k}$ (used as a novel input of the action predictor) and its language-aligned 3D keypoint $p_{t_k}$ (for cropping a region of interest, as usually done in coarse-to-fine policies). We discuss next how this prediction can be realized effectively.

Task decomposition enables reasoning about task plans before predicting actions. However, directly training the model $f_\theta$ to simultaneously generate a task plan $\mathcal{L}$ and predict step instruction $\ell_{t_k}$ and 3D keypoint $p_{t_k}$, given as inputs multi-view images obtained from observation $o_{t_{k-1}}$ and a high-level task description $L$, (i.e., $f_\theta(o_{t_{k-1}}, L) \rightarrow (\mathcal{L}, \ell_{t_k}, p_{t_k})$), does not ensure generalization to novel instructions. Previous studies (Kim et al., 2024; Intelligence et al., 2025; Zhao et al., 2025; Gao et al., 2025) indicate that VLAs exhibit a strong bias towards visual inputs, due to the richer information embedded in the visual inputs. This reliance often causes failures to follow novel language instructions. Under our data-scarce training setting, this issue is intensified by the limited diversity of language instructions. To address this issue, we propose decoupling task planning from keypoint prediction via a two-round inference protocol. First, a purely textual query generates a language plan, the sequence of all step instructions $\mathcal{L}$. Second, the visual inputs augmented with this plan are used to predict the corresponding step instruction and keypoint:

$$f_\theta(L_i) \rightarrow \mathcal{L}, f_\theta(o_{t_{k-1}}, \mathcal{L}) \rightarrow (\ell_{t_k}, p_{t_k}). \tag{2}$$

This approach not only mitigates visual bias but also enables training with an auxiliary dataset of language plans, which serves as a manual to enhance compositional task reasoning.

For a plan $\mathcal{L}$ corresponding to a task description $L$, directly fine-tuning a VLM to predict both the step instruction $\ell_{t_k}$ and its language-aligned 3D keypoint $p_{t_k}$ reveals to be insufficient due to the inadequate alignment of the visual-textual embeddings of the VLM for this 3D keypoint prediction. Instead, inspired by Chain-of-Thought reasoning (Mu et al., 2023; Zawalski et al., 2024; Zhao et al., 2025) for robotics, we design a reasoning process by training our model to first reason about the pixel positions of task-related objects, then generate a step instruction and finally predict the corresponding keypoint, changing the second step of Equation (2) to:

$$f_\theta(o_{t_{k-1}}, \mathcal{L}) \rightarrow (\boldsymbol{p}_{obj}, \ell_{t_k}, p_{t_k}), \tag{3}$$

where $\boldsymbol{p}_{obj}$ are the 3D positions of the task-related objects.

We further observe that for long-horizon, especially repetitive tasks (e.g., "stack several blocks"), only providing the entire task plan whether as input or output can degrade the performance. This often causes the model to generate repetitive sequences until reaching the output length limit or to struggle in determining the next step from an excessively long plan. To mitigate this, we introduce two ideas. First, we provide to the VLM an additional input: the step instruction predicted in the last timestep, which serves as a short-term memory cue to contextualize the current situation. Moreover, we decompose the plan into sub-tasks, i.e., $\mathcal{L} = \Big(\mathcal{L}_1 = (\ell_1, \ell_2, ..., \ell_n), \mathcal{L}_2 = (\ell_{n+1}, \ell_{n+2}, ...), ...\Big)$. For instance, "stack the blue and yellow cup on the red cup" is decomposed into two sequential sub-tasks: $\mathcal{L}_1$ "stack the blue on the red cup" followed by $\mathcal{L}_2$ "stack the yellow cup on the red cup". The model is trained to generate only the task plan of current sub-task, preventing repetition and improving focus:

$$f_\theta(L, \hat{\ell}) \rightarrow \mathcal{L}_m, f_\theta(o_{t_{k-1}}, \mathcal{L}_m, \hat{\ell}) \rightarrow (\boldsymbol{p}_{obj}, \ell_{t_k}, p_{t_k}), \tag{4}$$

where $\hat{\ell}$ is the step instruction predicted in the last timestep and $\mathcal{L}_m$ is the language plan of the $m^{th}$ sub-task. At the beginning of a trajectory, where no previous timestep exists, $\hat{\ell}$ is defined as "the robot is currently at the initial state" to indicate the initial state.

To further enhance the generalization ability of the coarse task planner to object variations, we include an auxiliary task of predicting the object positions. We randomly initialize diverse environments and record the RGB-D images of the scene along with the 3D positions of the objects. Following the same pre-processing, both the RGB-D images and the 3D positions of the objects are projected into canonical views. The projected multi-view images and pixel coordinates of the object positions in each view are used to construct an object position dataset. This dataset is then utilized to co-train the VLM, reinforcing its spatial understanding and improving zero-shot generalization to object variations.

## 4.2 FINE-GRAINED ACTION PREDICTOR

The fine-grained action predictor uses a predicted step instruction $\ell_{t_k}$, instead of the original high-level task description $L$, enabling more precise and generalizable skill learning. Considering the significant domain shift between the images focused around predicted $p_{t_k}$ from those used to pre-train standard VLMs, we decide to employ instead pre-trained specialized encoders to process these

inputs. Our feature encoding pipeline consists of three stages to construct a unified 3D-aware and language-aligned representation. First, RGB images and step instructions are processed through vision-language encoders to establish semantic alignment between visual and textual inputs. Second, depth images are encoded separately to extract explicit geometric structure. Finally, we generate a 3D position embedding from pixel-wise 3D coordinates to incorporate spatial awareness. These components are combined to form a 3D-aware, language-aligned representation for downstream fine-grained action prediction, following the architecture of prior work (Goyal et al., 2024).

## 5  EXPERIMENTS

We now present the experimental settings and results in simulation and real-world experiments.

### 5.1  SIMULATION RESULTS

**Experimental Set-up**   Our method is evaluated on GemBench (Garcia et al., 2025), a benchmark specifically designed for evaluating the generalization ability of a policy. A dataset containing 100 demonstrations per task along with a task description per trajectory is prepared for training. This training set contains 16 tasks with 31 variations. Within a trajectory, 4 cameras are placed at the front, left shoulder, right shoulder and wrist to collect RGB-D images as the observations. The resolution of the original RGB-D images is 256x256 while the resolution of the projected images is 224x224. Instead of evaluating on in-distribution tasks and variations, GemBench designs an evaluation set containing 4 levels of tasks, where different elements are varied:

  - **Placements (L1**: same 16 tasks (31 variations) as in training set, but with novel object placements.
  - **Rigid Objects (L2)**: 15 novel tasks (28 variations) with newly-colored or -shaped rigid objects.
  - **Articulated Objects (L3)**: 18 novel tasks (21 variations) with appearance or object variation.
  - **Long-horizon Tasks (L4)**: 6 novel long-horizon tasks (12 variations).

The specific configuration for tasks used for training and evaluation in GemBench are listed in Appendix A.1. We explain in detail how we create the language plan dataset and object position dataset in Appendix A.2. Following the evaluation setting in GemBench (Garcia et al., 2025), all trained models are evaluated with 20 episodes per task variation per seed, and 5 different seeds are used.

In our method, we finetune Qwen 2.5 VL-3B (Bai et al., 2025) as the coarse task planner. It is LoRA fine-tuned (Hu et al., 2022) with the object keypoint dataset, language plans, and robot trajectories. We use SigLIP (Zhai et al., 2023) to extract features from the RGB images and step instructions, leveraging its language-aligned representations. For depth images, we use DINOv2 (Oquab et al., 2023), which excels at capturing geometric structures like edges and contours. We further enhance these features by using the 3D coordinate of each pixel to construct a 3D position embedding. The hyperparameters, such as batch size and learning rate used in training are listed in Appendix A.3.

To construct the fine-tuning dataset from robot trajectories, we design a sampling strategy to choose samples from the trajectories. Apart from key-frame pairs of observation and action $(o_{t_k}, a_{t_{k+1}})$, RVT2 augments the training data by sampling observations every $n$ frames (e.g., every 10 frames). However, this results in an uneven number of samples per trajectory segment, due to the varying length of each segment. We initially attempted to sample observations within a window $(o_{t_k-m}, ..., o_{t_k}, ...o_{t_k+m})$ around the time step $t_k$ of the $k^{th}$ key-frame. However, observations at time steps before a key-frame and after a key-frame are visually similar while corresponding to distinct actions. For example, $o_{t_k-1}$ and $o_{t_k+1}$ are similar while their corresponding keypoints are the gripper positions at $t_k$ and $t_{k+1}$ respectively. Using all observations around the key-frames to fine-tune the VLM risks confusing it. Finally, we choose to use observations $(o_{t_k}, ...o_{t_k+m})$ at the time steps immediately following each key-frame. We empirically choose $m$ as 5 in all experiments.

**Main Results**   The evaluation results on GemBench are summarized in Table 1, reporting the average success rate for tasks at each generalization level. The detailed success rate for each task are recorded in Appendix A.4. The experimental results demonstrate the strong generalization ability of our method to novel tasks and object variations, as indicated by the performance gain on Level-2, Level-3, and Level-4 tasks. Our method achieves an overall success rate 12% higher than prior

| Models | Avg. Success ↑ | L1 | L2 | L3 | L4 |
|---|---|---|---|---|---|
| HiveFormer (Guhur et al., 2023) | 30.4 | 60.3 ± 1.5 | 26.1 ± 1.4 | 35.1 ± 1.7 | 0.0 ± 0.0 |
| PolarNet (Chen et al., 2023) | 38.4 | 77.7 ± 0.9 | 37.1 ± 1.4 | 38.5 ± 1.7 | 0.1 ± 0.2 |
| 3D Diffuser Actor (Ke et al., 2024) | 44.0 | 91.9 ± 0.8 | 43.4 ± 2.8 | 37.0 ± 2.2 | 0.0 ± 0.0 |
| RVT2 (Goyal et al., 2024) | 44.0 | 89.1 ± 0.8 | 51.0 ± 2.3 | 36.0 ± 2.2 | 0.0 ± 0.0 |
| 3D-LOTUS (Garcia et al., 2025) | 45.7 | **94.3** ± 1.4 | 49.9 ± 2.2 | 38.1 ± 1.1 | 0.3 ± 0.3 |
| 3D-LOTUS++ (Garcia et al., 2025) | 48.0 | 68.7 ± 0.6 | 64.5 ± 0.9 | 41.5 ± 1.8 | 17.4 ± 0.4 |
| BridgeVLA (Li et al., 2025b) | 50.0 | 91.1 ± 1.1 | 65.0 ± 1.3 | 43.8 ± 1.2 | 0.0 ± 0.0 |
| **CLAP (Proposed)** | **62.0** | 83.9 ± 0.3 | **83.2** ± 1.9 | **49.6** ± 2.1 | **31.4** ± 0.6 |

Table 1: **Multi-Task Performance on GemBench.** Here are the average success rates of 4 levels of evaluation tasks from Gembench. Except CLAP, we use the results reported in BridgeVLA.

| Exp ID | Language Plan Data | Object Keypoints Data | Last Step | Reason Plan | Reason Objects | Pretrained Encoder | Level 1 | Level 2 | Level 3 | Level 4 | Avg. Succ. |
|---|---|---|---|---|---|---|---|---|---|---|---|
| 1 | ✗ | ✗ | ✗ | ✗ | ✗ | ✓ | 86.9 | 68.2 | 36.4 | 0.4 | 48.0 |
| 2 | ✗ | ✗ | ✗ | ✗ | ✓ | ✓ | 81.8 | 74.8 | 39.0 | 0 | 48.9 |
| 3 | ✓ | ✗ | ✗ | ✓ | ✗ | ✓ | 83.4 | 66.1 | 41.9 | 2.0 | 48.3 |
| 4 | ✓ | ✗ | ✓ | ✓ | ✓ | ✓ | 84.8 | 81.4 | 43.8 | 30.4 | 60.1 |
| 5 | ✓ | ✓ | ✓ | ✓ | ✓ | ✗ | 82.4 | 79.1 | 44.5 | 25 | 57.8 |
| 6 | ✓ | ✓ | ✓ | ✓ | ✓ | ✓ | 83.8 | 83.2 | 49.6 | 31.4 | 62.0 |

Table 2: **Ablation study of CLAP on GemBench.** Here are the average success rates of 4 levels of evaluation tasks from Gembench under different training settings.

state-of-the-art method (Li et al., 2025b). Notably, this improvement is obtained using only 20 trajectories per task variation for training, significantly fewer than the 100 trajectories used by other baselines. Furthermore, our design leads to substantial performance gain on the most challenging Level-4 tasks, where several baselines methods fail consistently. We also include an experiments on the effects of using different number of trajectories in Appendix A.5.

**Ablation** We further experimentally validate the design choice for both coarse task planner and fine-grained action predictor on GemBench. The configurations are detailed below and corresponding results are presented in Table 2. A specific ablation on the inputs of the fine-grained action predictor is include in Appendix A.5.

1) **Base** In the base version (corresponding to Exp ID 1), the coarse task planner is trained with only the robot trajectories to predict step instruction and the corresponding keypoints. We use this version as a baseline to ablate our method.

2) **Object Reasoning** To adapt the pre-trained VLM for 3D keypoint prediction, we introduce a structured reasoning procedure where the model first localizes task-relevant objects before predicting the step instruction and its corresponding keypoint. We evaluate the efficacy of this object position reasoning in Exp ID 2. A comparison with the base model (Exp ID 1) reveals a performance improvement on Level-2 and Level-3 tasks, indicating enhanced generalization to object variations.

3) **Language Plan Reasoning** The proposed task decomposition enables a two-round conversation, where a language plan is first generated through textual reasoning, followed by keypoint prediction. This approach also permits the inclusion of additional language plans during training to enhance compositional generalization. Compared to the base model (Exp ID 1), this version shows improvements on Level 3 and Level 4 tasks, demonstrating stronger generalization to novel task variations.

4) **Include Previous Step Instruction** Previous step instruction is included in the input as a short-term memory to help contextualize the current status. This design yields performance gains across Levels 2, 3, and 4, with particularly notable improvements on long-horizon tasks in Level 4.

5) **CLAP w/o Pre-trained Encoder** An ablation study (comparing Exp ID 5 and Exp ID 6) on the coarse planner confirms that incorporating the 3D-aware representation contributes to performance gains at all generalization levels.

6) **CLAP** Our method integrates all components mentioned above. A comparison between Exp ID 4 and ours can further validate the performance gain from adding the object position dataset.

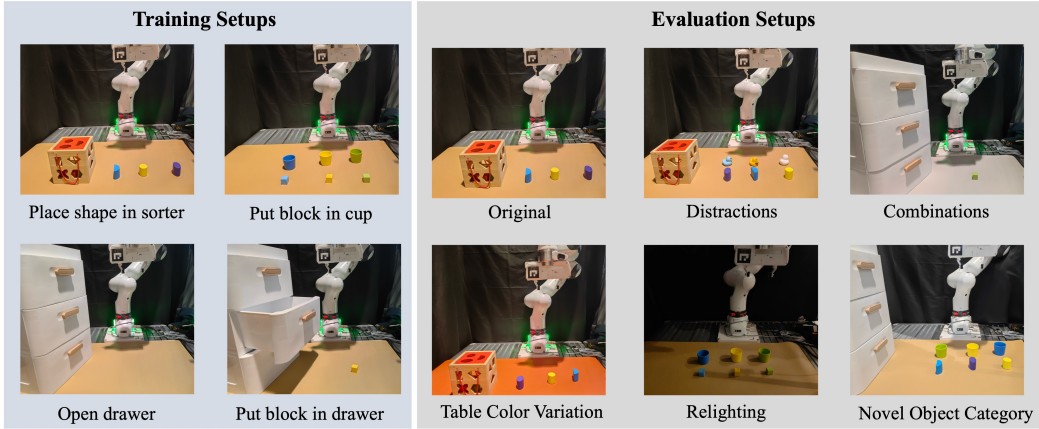

Figure 3: **Overview of the tasks in real-world experiments.** There are four training tasks: put shape in shape sorter, put block in cup, open drawer, put block in drawer. We evaluate the same tasks under different visual perturbations and novel tasks designed based on the training tasks.

| | No Variation | | Table Color | | Distracted Objects | | Light Strength | | Average Succ. | |
|---|---|---|---|---|---|---|---|---|---|---|
| | RVT2 | CLAP | RVT2 | CLAP | RVT2 | CLAP | RVT2 | CLAP | RVT2 | CLAP |
| place shape in shape sorter | 60% | **60%** | 35% | **50%** | 30% | **40%** | 20% | **50%** | 36.2% | **50%** |
| put block in cup with same color | 40% | **100%** | 20% | **70%** | 20% | **80%** | 20% | **80%** | 25% | **82.5%** |
| open drawer | 100% | **100%** | 85% | **95%** | 100% | **100%** | 0% | **100%** | 71.2% | **98.7%** |
| put block in open drawer | 40% | **90%** | 25% | **90%** | 20% | **70%** | 0% | **90%** | 21.2% | **85%** |
| put block in cup with different color | 20% | **100%** | 15% | **70%** | 0% | **70%** | 10% | **80%** | 11.2% | **80%** |
| put shape in open drawer | 30% | **80%** | 20% | **65%** | 10% | **70%** | 0% | **80%** | 15% | **73.7%** |
| put shape in cup | 0% | **80%** | 0% | **70%** | 0% | **60%** | 0% | **70%** | 0% | **70%** |
| put block in close drawer | 0% | **90%** | 0% | **75%** | 0% | **70%** | 0% | **80%** | 0% | **78.7%** |
| average success rate | 36.2% | **87.5%** | 25% | **73.1%** | 22.5% | **70%** | 6.2% | **78.7%** | 22.5% | **77.3%** |

Table 3: **Real-world Performance.** Here are the average success rate under different generalization settings for real-world experiments.

## 5.2 REAL-WORLD EXPERIMENTS

**Experimental Setting** We keep the training settings the same as in the simulation and list key modifications here. In the real-world experiments, we use a single camera (Intel RealSense D435i) to collect RGB-D images of size 640x360. 10 trajectories are collected per task to cover all variations of each task. The hyperparameters used for training the models are listed in Appendix A.3. The training tasks, illustrated in Figure 3, are listed below. 1) Place shape in shape sorter: insert objects into a box with 3 variations on the object shape. 2) Put block in cup: put a colored block in a same-colored cup with 3 variations on colors. 3) Open drawer: open a drawer with 3 variations on the handles. 4) Put a block in drawer: put a colored block in an open drawer with 3 variations on colors.

We assess the generalization ability of CLAP along two key dimensions (see Appendix A.6 for an overview of these evaluation tasks):

1) **Visual Perturbations**: The model is tested on the tasks same as the training tasks but with the following conditions: different table colors, introducing distracting objects and altered backgrounds.

2) **Task/Object Variations**: Generalization is evaluated through: i) Object substitution (e.g., placing a "shape" object into a cup), and ii) Skill composition (e.g., combining "open drawer" and "place block" into a single, sequential task).

**Results** The results of evaluating the trained models in real-world experiments are summarized in Table 3. Our method achieves a strong generalization ability to novel tasks and object variations, trained with only 10 demonstrations per task. CLAP achieves 54.8% higher average success rates compared to RVT2 on the evaluation tasks.

## 6 CONCLUSIONS

We propose a novel coarse-to-fine 3D manipulation policy, where tasks are decomposed and pre-trained models are leveraged in the hierarchical architecture. Our method demonstrates strong generalization capabilities while maintaining the sample efficiency inherent to coarse-to-fine approaches. Although leveraging pre-trained models for robotics tasks is common, their effective adaptation for generalizable and precise is still under-explored. We hope this work inspires further research into building highly generalizable and sample-efficient 3D manipulation policies. Our method has two key limitations. First, key-frame based imitation learning is suitable for structured tasks that can be easily decomposed into discrete steps. Unstructured tasks, such as wiping a desk, where key-frames are difficult to define, present a significant challenge. Moreover, the current framework lacks a robust error-correction mechanism. An incorrect action prediction at any step might lead to task failure. A promising future direction is to integrate a self-correction module to enhance robustness.

**Acknowledgement** This work has been supported in part by the program of National Natural Science Foundation of China (No. 62176154), the program of National Natural Science Foundation of China (No. 62503322), Shanghai Magnolia Funding Pujiang Program (No. 23PJ1404400), and the AI for Science Seed Program of Shanghai Jiao Tong University (project number 2025AI4S-QY06).

**LLM Usage** We used Deepseek (AI, 2024) and ChatGPT (OpenAI, 2024) for grammar check and related work retrieval. The authors have reviewed the content generated by the LLM.

**Ethics Statement** We adhere to the ICLR Code of Ethics and take full responsibility for the final content.

**Reproducibility Statement** To ensure reproducibility, we provide a comprehensive description of our method and experimental setup in the Section 4 and the Section 5, document all hyperparameters in the appendix Table 6, and release our code at `https://github.com/Jianshu-Hu/Generalizable-CLAP`.

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

Table 4: **Number of samples.** We record the number of samples in training set and validation set for different datasets used in the simulation experiments for GemBench.

|  | Language Plan | Object Position | Robot Trajectory Data |
|---|---|---|---|
| Train | 30515 | 39777 | 169665 |
| Validation | 6103 | 16650 | 17400 |

## A  APPENDIX

### A.1  GEMBENCH TASK SPECIFICATION

The detailed tasks and variations in GemBench (Garcia et al., 2025) used for training and evaluation are listed in Table 5.

### A.2  TRAINING DATA SPECIFICATION

We have three datasets in total for finetuning the pre-trained VLM: a language plan dataset, a object position dataset and a robot trajectory dataset.

- For language plan dataset, we label each task with a language plan according to the decomposition of the task. Note that one task may contain different variations. Each variation of the task is assigned with a different language plan. We augment the data by asking LLM to provide some variations on the language descriptions of the plan.
- For object position dataset, we randomly initialize the scene of the tasks and apply transformations (translation and rotation) on the scene. We record the RGB-D images along with the 3D positions of the objects. The 3D positions of the objects are projected into canonical views to obtain the pixel positions in each view.
- For robot trajectory dataset, we explain in the Section 5.1 on how we sample transitions from the demonstrations to build observation-action pair. Transformations (translation and rotation) are also applied on the transitions to augment the data.

In practice, the number of training samples for each dataset used for GemBench is listed in Table 4.

### A.3  EXPERIMENTAL DETAILS

All experiments are conducted on 4 NVIDIA RTX 4090 GPU. The hyperparameters and training time are listed in Table 6.

### A.4  GEMBENCH SUCCESS RATE PER TASK

The results for each task across different generalization levels in Gembench are listed in Table 7, Table 8, Table 9, Table 10.

### A.5  ADDITIONAL ABLATION STUDY

We first show the performance of our method on GemBench trained with different numbers of robot trajectories (10, 20, 50, 100) in Table 11. Further increasing the number of robot trajectory improves on the in-domain performance (L1) while does not help in the average success rate. We conclude that the data efficiency is attributed to the following points.

- Task decomposition allows the agent to learn common skills among different tasks and allows compositional generalization.
- Bridging VLM and 3D manipulation with projected images, separating task planning from keypoint prediction, and Chain-of-Thought reasoning lead to an effective adaptation of pre-trained VLM to the coarse task planner.
- Data efficiency with respect to the robot trajectories also benefits from leveraging language plans and object position data. By co-training on these additional sources, our framework

Table 5: **Training and evaluation tasks & variations in GemBench.** The evaluation tasks contain four levels of generalization, where Level 1 evaluates the generalization to novel placements, Level 2 novel rigid objects, Level 3 novel articulated objects, and Level 4 novel long-horizon tasks.

| | Train / Level 1 | | Level 2 | | Level 3 | | | Level 4 |
|---|---|---|---|---|---|---|---|---|
| | Task | Variation | Color | Shape | Instance | Category | Action-Part | Long-horizon |
| Press | Push button | maroon button
navy button
yellow button | azure button
rose button
white button | Lamp on | | | | 2 buttons
3 buttons
4 buttons |
| Pick | Pick and lift | red block
lime block
cyan block
magenta cup | teal block
violet block
black block
gray cup | red cylinder
red star
red moon | | | | |
| | Pick up cup | silver cup
orange cup | olive cup
purple cup | red toy | | | | |
| Push | Slide block | green target
blue target | pink target
yellow target | | | | | |
| | Reach and drag | teal target
black target | cyan target
navy target | | | | | |
| Screw | Close jar | azure jar
violet jar | blue jar
green jar | | | | | |
| | Screw bulb | rose bulb
white bulb | lime bulb
maroon bulb | | | | | |
| Close | Close fridge
Close laptop lid
Close microwave | fridge
laptop lid
microwave | | | fridge2
laptop lid2
microwave2 | grill | door
box
drawer | |
| Open | Open door
Open box | door
box | | | door2
box2
drawer2, drawer3
long drawer w/ 4 levels | toilet | fridge
laptop lid
microwave
middle drawer | Take shoes
out of box
Put 3 items
in drawer |
| | Open drawer | bottom drawer
top drawer | | | | | | |
| Put/
Stack | Stack blocks | 2 gray blocks
2 olive blocks
2 purple blocks | 2 orange blocks
2 silver blocks
2 magenta blocks | | | | | Stack 3-4
blocks
Stack 2 cups |
| | Put groceries | crackers box
soup can | | mustard bottle
sugar box | | | | Put all
groceries |
| | Put money | bottom shelf
middle shelf | | Put cube in
bottom shelf | | | top shelf | |

Table 6: **Training time and hyperparameters used in different experiments.** Here we list the training time and hyperparameters used for training the model with GemBench and real-world data.

| | GemBench | | Real-world | |
|---|---|---|---|---|
| | Coarse task planner | Fine-grained Action Predictor | Coarse task planner | Fine-grained Action Predictor |
| Training time | 6 hours | 3 hours | 1 hour | 1 hour |
| Learning rata | 3e-4 | 0.0024 | 1e-4 | 0.0024 |
| Batch Size | 64 | 192 | 64 | 192 |
| Epochs | 1 | 5 | 1 | 3 |
| Lora Rank | 8 | / | 8 | / |
| Lora Alpha | 32 | / | 32 | / |
| Freeze Vit | False | / | False | / |
| Freeze Aligner | True | / | True | / |
| Freeze LLM | False | / | False | / |

equips the model with task-planning and object-grounding abilities, leaving it to learn only how to act given sub-task instructions and known object locations. In contrast, prior approaches required the model to learn task planning, object detection, and action prediction solely from robot trajectories.

We then show the results of variations on inputs to the fine-grained action predictor in Table 12. Removing some components in the fine-grained action predictor leads to performance drop to some extend.

## A.6 REAL-WORLD EXPERIMENTS

An overview of the tasks used in real-world experiments are shown in Figure 4.

| Method | Avg. | Close Fridge+0 | Close Jar+15 | Close Jar+16 | CloseLaptop Lid+0 | Close Microwave+0 | LightBulb In+17 | LightBulb In+19 | Open Box+0 | Open Door+0 | Open Drawer+0 |
|---|---|---|---|---|---|---|---|---|---|---|---|
| HiveFormer (Guhur et al., 2023) | 60.3±1.5 | 96±4.2 | 64±13.9 | 92±2.7 | 90±3.5 | 88±7.6 | 12±4.5 | 13±6.7 | 4±4.2 | 53±15.2 | 15±12.2 |
| PolarNet (Chen et al., 2023) | 77.6±0.9 | 99±2.2 | 99±2.2 | 99±2.2 | 95±3.5 | 98±2.7 | 72±12.5 | 71±6.5 | 32±11.5 | 69±8.9 | 61±12.4 |
| 3D Diffuser Actor (Ke et al., 2024) | 91.9±0.8 | 100±0.0 | 100±0.0 | 100±0.0 | 99±2.2 | 100±0.0 | 85±5.0 | 88±2.7 | 11±2.2 | 96±4.2 | 82±9.1 |
| RVT2 (Goyal et al., 2024) | 89.0±0.8 | 77±11.0 | 97±4.5 | 88±5.7 | 77±13.0 | 100±0.0 | 93±5.7 | 91±8.2 | 7±4.5 | 98±4.5 | 93±5.7 |
| 3D-LOTUS (Garcia et al., 2025) | 94.3±3.5 | 96±3.7 | 100±0.0 | 100±0.0 | 98±2.5 | 98±4.0 | 84±7.4 | 85±9.5 | 99±2.0 | 77±2.5 | 83±8.7 |
| 3D-LOTUS++ (Garcia et al., 2025) | 68.7±0.6 | 95±0.0 | 100±0.0 | 99±2.0 | 28±2.5 | 87±5.1 | 55±10.5 | 45±8.9 | 55±8.9 | 79±9.7 | 68±12.5 |
| BridgeVLA (Li et al., 2025b) | 91.1±1.1 | 99±2.0 | 98±4.0 | 100±0.0 | 97±2.5 | 85±5.5 | 90±5.5 | 87±7.5 | 76±10.2 | 70±12.3 | 86±5.8 |
| CLAP | 83.9±0.3 | 88±4.5 | 98±2.7 | 100±0.0 | 88±9.1 | 99±2.2 | 84±6.5 | 76±9.1 | 17±5.7 | 82±4.5 | 87±13.0 |

| Method | Open Drawer+2 | Pick& Lift+1 | Pick& Lift+2 | Pick& Lift+7 | PickUp Cup+8 | PickUp Cup+9 | PickUp Cup+11 | Push Button+0 | Push Button+3 | Push Button+4 | PutIn Cupboard+0 |
|---|---|---|---|---|---|---|---|---|---|---|---|
| HiveFormer (Guhur et al., 2023) | 59±7.4 | 86±4.2 | 92±6.7 | 93±2.7 | 83±7.6 | 69±12.9 | 61±19.8 | 84±11.9 | 68±6.7 | 87±7.6 | 34±8.2 |
| PolarNet (Chen et al., 2023) | 90±7.1 | 92±9.1 | 84±7.4 | 88±5.7 | 82±7.6 | 79±4.2 | 72±10.4 | 100±0.0 | 100±0.0 | 99±2.2 | 52±7.6 |
| 3D Diffuser Actor (Ke et al., 2024) | 97±4.5 | 99±2.2 | 99±2.2 | 99±2.2 | 96±2.2 | 97±4.5 | 98±2.7 | 98±2.7 | 96±4.2 | 98±2.7 | 85±5.0 |
| RVT2 (Goyal et al., 2024) | 94±4.2 | 99±2.2 | 98±2.7 | 100±0.0 | 99±2.2 | 99±2.2 | 99±2.2 | 100±0.0 | 100±0.0 | 100±0.0 | 88±8.4 |
| 3D-LOTUS (Garcia et al., 2025) | 93±6.0 | 99±2.0 | 100±0.0 | 99±2.0 | 97±4.0 | 96±3.7 | 94±4.9 | 99±2.0 | 99±2.0 | 100±0.0 | 89±5.8 |
| 3D-LOTUS++ (Garcia et al., 2025) | 75±4.5 | 97±6.0 | 94±3.7 | 93±5.1 | 86±8.0 | 88±6.8 | 91±4.9 | 100±0.0 | 100±0.0 | 100±0.0 | 1±2.0 |
| BridgeVLA (Li et al., 2025b) | 99±2.0 | 99±2.0 | 100±0.0 | 98±2.5 | 96±2.0 | 94±3.7 | 99±2.0 | 100±0.0 | 98±4.0 | 98±4.0 | 74±6.6 |
| CLAP | 98±2.7 | 98±2.7 | 99±2.2 | 99±2.2 | 94±5.5 | 99±2.2 | 93±7.6 | 100±0.0 | 100±0.0 | 97±2.7 | 58±10.4 |

| Method | PutIn Cupboard+3 | PutMoney InSafe+0 | PutMoney InSafe+1 | Reach& Drag+14 | Reach& Drag+18 | Slide Block+0 | Slide Block+1 | Stack Blocks+30 | Stack Blocks+36 | Stack Blocks+39 |
|---|---|---|---|---|---|---|---|---|---|---|
| HiveFormer (Guhur et al., 2023) | 74±6.5 | 85±3.5 | 88±2.7 | 37±5.7 | 32±7.6 | 99±2.2 | 91±12.4 | 6±5.5 | 7±4.5 | 6±4.2 |
| PolarNet (Chen et al., 2023) | 88±4.5 | 93±4.5 | 95±5.0 | 99±2.2 | 99±2.2 | 100±0.0 | 0±0.0 | 34±10.8 | 30±9.4 | 36±12.9 |
| 3D Diffuser Actor (Ke et al., 2024) | 82±11.5 | 95±5.0 | 98±2.7 | 100±0.0 | 99±2.2 | 100±0.0 | 89±4.2 | 88±7.6 | 85±6.1 | 89±5.5 |
| RVT2 (Goyal et al., 2024) | 80±6.1 | 93±8.4 | 96±8.5 | 85±10.0 | 94±2.2 | 100±0.0 | 37±6.7 | 88±5.7 | 93±2.7 | 88±11.5 |
| 3D-LOTUS (Garcia et al., 2025) | 72±11.2 | 94±3.7 | 99±2.0 | 99±2.0 | 100±0.0 | 100±0.0 | 100±0.0 | 94±5.8 | 91±6.6 | 90±4.5 |
| 3D-LOTUS++ (Garcia et al., 2025) | 2±2.5 | 22±6.8 | 16±4.9 | 94±3.7 | 62±8.7 | 100±0.0 | 65±5.5 | 86±5.8 | 20±4.5 | 28±13.6 |
| BridgeVLA (Li et al., 2025b) | 84±6.6 | 79±9.7 | 86±3.7 | 96±5.8 | 97±4.0 | 100±0.0 | 90±5.5 | 77±8.1 | 87±4.0 | 85±7.8 |
| CLAP | 69±12.4 | 80±6.1 | 82±7.6 | 90±3.5 | 90±34.5 | 55±5.0 | 5±5.0 | 96±4.2 | 85±3.5 | 90±6.1 |

Table 7: **Per-task Success Rate on GemBench Level 1.**

| Method | Avg. | Push Button+13 | Push Button+15 | Push Button+17 | Pick& Lift+14 | Pick& Lift+16 | Pick& Lift+18 | PickUp Cup+10 | PickUp Cup+12 | PickUp Cup+13 |
|---|---|---|---|---|---|---|---|---|---|---|
| HiveFormer (Guhur et al., 2023) | 26.1±1.4 | 97±2.7 | 85±10.0 | 88±2.7 | 21±6.5 | 9±4.2 | 8±6.7 | 30±7.1 | 22±13.5 | 26±10.6 |
| PolarNet (Chen et al., 2023) | 37.1±1.4 | 100±0.0 | 100±0.0 | 85±7.9 | 3±4.5 | 1±2.2 | 0±0.0 | 48±5.5 | 46±8.9 | 16±6.5 |
| 3D Diffuser Actor (Ke et al., 2024) | 43.4±2.8 | 87±13.0 | 81±6.5 | 60±9.4 | 9±4.2 | 18±9.1 | 0±0.0 | 84±5.5 | 60±11.7 | 62±13.0 |
| RVT2 (Goyal et al., 2024) | 51.0±2.3 | 100±0.0 | 100±0.0 | 100±0.0 | 47±7.6 | 29±9.6 | 8±4.5 | 81±8.2 | 59±9.6 | 72±9.7 |
| 3D-LOTUS (Garcia et al., 2025) | 49.9±2.2 | 99±2.0 | 100±0.0 | 100±0.0 | 3±2.5 | 18±8.7 | 33±9.3 | 89±3.7 | 78±8.7 | 57±7.5 |
| 3D-LOTUS++ (Garcia et al., 2025) | 64.5±0.9 | 99±2.0 | 100±0.0 | 99±2.0 | 94±3.7 | 96±3.7 | 95±3.2 | 79±4.9 | 89±9.7 | 84±10.2 |
| BridgeVLA (Li et al., 2025b) | 65.0±1.3 | 100±0.0 | 100±0.0 | 100±0.0 | 74±9.7 | 89±4.9 | 0±0.0 | 91±3.7 | 90±3.2 | 90±6.3 |
| CLAP | 83.2±1.9 | 100±0.0 | 100±0.0 | 100±0.0 | 99±2.2 | 100±0.0 | 98±2.7 | 93±4.5 | 97±2.7 | 98±2.7 |

| Method | Stack Blocks+24 | Stack Blocks+27 | Stack Blocks+33 | Slide Block+2 | Slide Block+3 | Close Jar+3 | Close Jar+4 | LightBulb In+1 | LightBulb In+2 | Lamp On+0 |
|---|---|---|---|---|---|---|---|---|---|---|
| HiveFormer (Guhur et al., 2023) | 0±0.0 | 4±4.2 | 0±0.0 | 0±0.0 | 0±0.0 | 0±0.0 | 0±0.0 | 4±4.2 | 0±0.0 | 7±4.5 |
| PolarNet (Chen et al., 2023) | 1±2.2 | 2±2.7 | 6±8.2 | 0±0.0 | 0±0.0 | 20±10.6 | 82±5.7 | 22±11.5 | 17±8.4 | 14±10.8 |
| 3D Diffuser Actor (Ke et al., 2024) | 66±13.9 | 82±2.7 | 50±14.6 | 0±0.0 | 0±0.0 | 23±16.8 | 82±5.7 | 51±17.8 | 60±10.0 | 7±7.6 |
| RVT2 (Goyal et al., 2024) | 18±4.5 | 56±16.7 | 45±13.7 | 0±0.0 | 1±2.2 | 7±7.6 | 77±5.7 | 68±14.4 | 6±6.5 | 0±0.0 |
| 3D-LOTUS (Garcia et al., 2025) | 13±8.1 | 40±9.5 | 69±5.8 | 0±0.0 | 0±0.0 | 71±5.8 | 90±4.5 | 24±4.9 | 41±8.6 | 0±0.0 |
| 3D-LOTUS++ (Garcia et al., 2025) | 22±9.3 | 83±7.5 | 59±3.7 | 27±9.8 | 5±3.2 | 98±2.5 | 96±3.7 | 56±9.7 | 43±7.5 | 2±2.0 |
| BridgeVLA (Li et al., 2025b) | 61±10.7 | 51±13.2 | 79±8.6 | 12±9.3 | 3±4.0 | 66±6.6 | 88±4.0 | 66±8.6 | 74±5.8 | 7±4.0 |
| CLAP | 95±3.5 | 86±2.2 | 91±4.2 | 18±5.7 | 68±5.7 | 95±3.5 | 98±2.7 | 66±5.5 | 81±6.5 | 20±6.1 |

| Method | Reach& Drag+5 | Reach& Drag+7 | PutCube InSafe+0 | Pick&Lift Cylinder+0 | Pick&Lift Star+0 | Pick&Lift Moon+0 | Pick&Lift Toy+0 | PutIn Cupboard+7 | PutIn Cupboard+8 |
|---|---|---|---|---|---|---|---|---|---|
| HiveFormer (Guhur et al., 2023) | 1±2.2 | 0±0.0 | 4±2.2 | 78±5.7 | 73±7.6 | 88±2.7 | 87±4.5 | 0±0.0 | 0±0.0 |
| PolarNet (Chen et al., 2023) | 61±8.2 | 10±6.1 | 40±14.1 | 93±6.7 | 88±8.4 | 93±6.7 | 90±3.5 | 0±0.0 | 0±0.0 |
| 3D Diffuser Actor (Ke et al., 2024) | 0±0.0 | 64±6.5 | 3±2.7 | 99±2.2 | 43±17.9 | 91±9.6 | 30±9.4 | 0±0.0 | 3±4.5 |
| RVT2 (Goyal et al., 2024) | 91±2.2 | 89±6.5 | 6±5.5 | 98±2.7 | 98±4.5 | 94±4.2 | 78±8.4 | 0±0.0 | 0±0.0 |
| 3D-LOTUS (Garcia et al., 2025) | 95±4.5 | 18±10.8 | 25±5.5 | 88±8.7 | 69±6.6 | 80±8.4 | 96±3.7 | 0±0.0 | 0±0.0 |
| 3D-LOTUS++ (Garcia et al., 2025) | 94±2.0 | 64±12.4 | 37±5.1 | 91±2.0 | 94±3.7 | 29±6.6 | 71±2.0 | 1±2.0 | 0±0.0 |
| BridgeVLA (Li et al., 2025b) | 94±3.7 | 96±3.7 | 3±2.5 | 98±2.5 | 99±2.0 | 95±3.2 | 93±5.1 | 0±0.0 | 0±0.0 |
| CLAP | 95±3.5 | 90±6.1 | 61±14.7 | 97±2.7 | 100±0.0 | 98±2.7 | 84±8.2 | 50±11.2 | 53±12.5 |

Table 8: **Per-task Success Rate on GemBench Level 2.**

**Examples of Real-world Tasks**

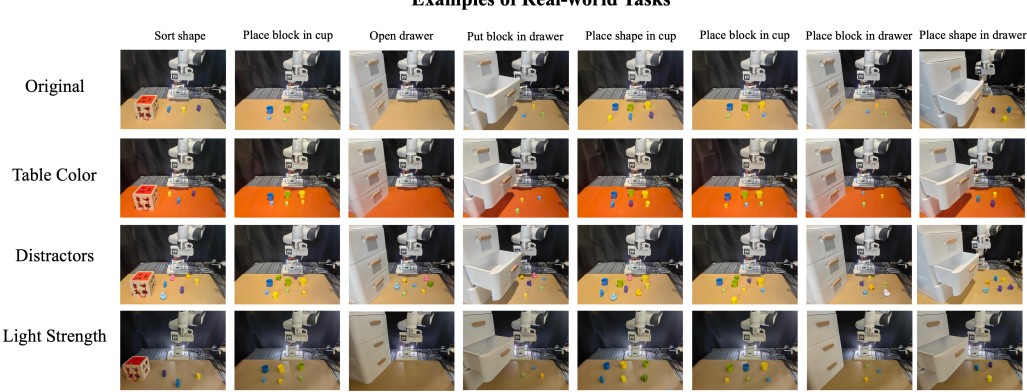

Figure 4: **Overview of the evaluation tasks in real-world experiments.** We evaluate the all these eight tasks acroos different variations and record the success rate.

| Method | Avg. | Close Door+0 | Close Box+0 | Close Fridge2+0 | CloseLaptop Lid2+0 | Close Microwave2+0 | Open Door2+0 | Open Box2+0 |
|---|---|---|---|---|---|---|---|---|
| HiveFormer (Guhur et al., 2023) | $35.1_{\pm1.7}$ | $0_{\pm0.0}$ | $1_{\pm2.2}$ | $34_{\pm9.6}$ | $52_{\pm9.1}$ | $15_{\pm7.1}$ | $32_{\pm11.5}$ | $5_{\pm3.5}$ |
| PolarNet (Chen et al., 2023) | $38.5_{\pm1.7}$ | $0_{\pm0.0}$ | $0_{\pm0.0}$ | $78_{\pm5.7}$ | $26_{\pm8.2}$ | $74_{\pm6.5}$ | $33_{\pm6.7}$ | $23_{\pm8.4}$ |
| 3D Diffuser Actor (Ke et al., 2024) | $37.0_{\pm2.2}$ | $0_{\pm0.0}$ | $0_{\pm0.0}$ | $97_{\pm2.7}$ | $23_{\pm6.7}$ | $88_{\pm7.6}$ | $\mathbf{86}_{\pm7.4}$ | $67_{\pm9.8}$ |
| RVT2 (Goyal et al., 2024) | $36.0_{\pm2.2}$ | $1_{\pm2.2}$ | $2_{\pm2.7}$ | $72_{\pm6.7}$ | $42_{\pm14.0}$ | $71_{\pm8.9}$ | $79_{\pm6.5}$ | $5_{\pm6.1}$ |
| 3D-LOTUS (Garcia et al., 2025) | $38.1_{\pm1.1}$ | $0_{\pm0.0}$ | $\mathbf{58}_{\pm8.1}$ | $36_{\pm9.7}$ | $54_{\pm10.7}$ | $85_{\pm7.1}$ | $42_{\pm6.8}$ | $11_{\pm6.6}$ |
| 3D-LOTUS++ (Garcia et al., 2025) | $41.5_{\pm1.8}$ | $1_{\pm2.0}$ | $29_{\pm8.6}$ | $93_{\pm2.5}$ | $50_{\pm9.5}$ | $\mathbf{99}_{\pm2.0}$ | $52_{\pm10.3}$ | $16_{\pm8.0}$ |
| BridgeVLA (Li et al., 2025b) | $\mathbf{43.8}_{\pm1.2}$ | $0_{\pm0.0}$ | $1_{\pm2.0}$ | $95_{\pm5.5}$ | $\mathbf{77}_{\pm4.0}$ | $54_{\pm10.2}$ | $68_{\pm10.8}$ | $\mathbf{74}_{\pm4.9}$ |
| CLAP | $49.6_{\pm2.1}$ | $\mathbf{3}_{\pm2.7}$ | $9_{\pm5.5}$ | $92_{\pm4.5}$ | $35_{\pm9.4}$ | $79_{\pm5.5}$ | $56_{\pm6.5}$ | $1_{\pm2.2}$ |

| Method | Open Drawer2+0 | Open Drawer3+0 | OpenDrawer Long+0 | OpenDrawer Long+1 | OpenDrawer Long+2 | OpenDrawer Long+3 | Toilet SeatUp+0 | Open Fridge+0 |
|---|---|---|---|---|---|---|---|---|
| HiveFormer (Guhur et al., 2023) | $59_{\pm11.9}$ | $39_{\pm11.9}$ | $78_{\pm8.4}$ | $82_{\pm4.5}$ | $49_{\pm4.2}$ | $57_{\pm11.5}$ | $6_{\pm4.2}$ | $0_{\pm0.0}$ |
| PolarNet (Chen et al., 2023) | $\mathbf{91}_{\pm4.2}$ | $29_{\pm8.2}$ | $84_{\pm11.9}$ | $88_{\pm5.7}$ | $\mathbf{63}_{\pm8.4}$ | $37_{\pm7.6}$ | $2_{\pm2.7}$ | $4_{\pm2.2}$ |
| 3D Diffuser Actor (Ke et al., 2024) | $19_{\pm8.2}$ | $1_{\pm2.2}$ | $15_{\pm5.0}$ | $35_{\pm13.7}$ | $26_{\pm9.6}$ | $79_{\pm12.9}$ | $0_{\pm0.0}$ | $7_{\pm5.7}$ |
| RVT2 (Goyal et al., 2024) | $81_{\pm11.9}$ | $0_{\pm0.0}$ | $84_{\pm8.2}$ | $39_{\pm10.8}$ | $11_{\pm8.9}$ | $75_{\pm6.1}$ | $7_{\pm5.7}$ | $0_{\pm0.0}$ |
| 3D-LOTUS (Garcia et al., 2025) | $90_{\pm3.2}$ | $22_{\pm8.1}$ | $56_{\pm13.9}$ | $33_{\pm11.2}$ | $17_{\pm8.1}$ | $75_{\pm6.3}$ | $0_{\pm0.0}$ | $4_{\pm5.8}$ |
| 3D-LOTUS++ (Garcia et al., 2025) | $70_{\pm5.5}$ | $41_{\pm4.9}$ | $72_{\pm4.0}$ | $52_{\pm10.8}$ | $23_{\pm8.1}$ | $78_{\pm5.1}$ | $\mathbf{8}_{\pm5.1}$ | $0_{\pm0.0}$ |
| BridgeVLA (Li et al., 2025b) | $65_{\pm6.3}$ | $\mathbf{87}_{\pm6.0}$ | $59_{\pm8.6}$ | $34_{\pm8.0}$ | $18_{\pm10.3}$ | $\mathbf{85}_{\pm8.4}$ | $6_{\pm5.8}$ | $7_{\pm2.5}$ |
| CLAP | $68_{\pm8.4}$ | $\mathbf{87}_{\pm7.6}$ | $44_{\pm10.8}$ | $\mathbf{94}_{\pm5.5}$ | $14_{\pm5.5}$ | $76_{\pm13.4}$ | $7_{\pm4.5}$ | $3_{\pm4.5}$ |

| Method | OpenLaptop Lid+0 | Open Microwave+0 | PutMoney InSafe+2 | Open Drawer+1 | Close Drawer+0 | Close Grill+0 |
|---|---|---|---|---|---|---|
| HiveFormer (Guhur et al., 2023) | $\mathbf{100}_{\pm0.0}$ | $0_{\pm0.0}$ | $0_{\pm0.0}$ | $0_{\pm0.0}$ | $83_{\pm5.7}$ | $44_{\pm10.8}$ |
| PolarNet (Chen et al., 2023) | $\mathbf{100}_{\pm0.0}$ | $0_{\pm0.0}$ | $1_{\pm2.2}$ | $4_{\pm4.2}$ | $29_{\pm11.9}$ | $42_{\pm11.5}$ |
| 3D Diffuser Actor (Ke et al., 2024) | $\mathbf{100}_{\pm0.0}$ | $0_{\pm0.0}$ | $2_{\pm4.5}$ | $0_{\pm0.0}$ | $66_{\pm7.4}$ | $\mathbf{65}_{\pm13.7}$ |
| RVT2 (Goyal et al., 2024) | $93_{\pm5.7}$ | $0_{\pm0.0}$ | $0_{\pm0.0}$ | $6_{\pm2.2}$ | $78_{\pm8.4}$ | $9_{\pm4.2}$ |
| 3D-LOTUS (Garcia et al., 2025) | $\mathbf{100}_{\pm0.0}$ | $0_{\pm0.0}$ | $0_{\pm0.0}$ | $0_{\pm0.0}$ | $\mathbf{87}_{\pm8.1}$ | $29_{\pm6.6}$ |
| 3D-LOTUS++ (Garcia et al., 2025) | $86_{\pm6.6}$ | $0_{\pm0.0}$ | $13_{\pm8.1}$ | $0_{\pm0.0}$ | $69_{\pm5.8}$ | $19_{\pm13.9}$ |
| BridgeVLA (Li et al., 2025b) | $95_{\pm0.0}$ | $0_{\pm0.0}$ | $2_{\pm2.5}$ | $0_{\pm0.0}$ | $58_{\pm12.9}$ | $35_{\pm12.3}$ |
| CLAP | $78_{\pm9.1}$ | $0_{\pm0.0}$ | $\mathbf{76}_{\pm5.5}$ | $\mathbf{96}_{\pm6.5}$ | $84_{\pm8.2}$ | $40_{\pm7.1}$ |

Table 9: **Per-task Success Rate on GemBench Level 3.**

| Method | Avg. | Push Buttons4+1 | Push Buttons4+2 | Push Buttons4+3 | TakeShoes OutOfBox+0 | PutItems InDrawer+0 | PutItems InDrawer+2 |
|---|---|---|---|---|---|---|---|
| HiveFormer (Guhur et al., 2023) | $0_{\pm0.0}$ | $0_{\pm0.0}$ | $0_{\pm0.0}$ | $0_{\pm0.0}$ | $0_{\pm0.0}$ | $0_{\pm0.0}$ | $0_{\pm0.0}$ |
| PolarNet (Chen et al., 2023) | $0.1_{\pm0.2}$ | $1_{\pm2.2}$ | $0_{\pm0.0}$ | $0_{\pm0.0}$ | $0_{\pm0.0}$ | $0_{\pm0.0}$ | $0_{\pm0.0}$ |
| 3D Diffuser Actor (Ke et al., 2024) | $0_{\pm0.0}$ | $0_{\pm0.0}$ | $0_{\pm0.0}$ | $0_{\pm0.0}$ | $0_{\pm0.0}$ | $0_{\pm0.0}$ | $0_{\pm0.0}$ |
| RVT2 (Goyal et al., 2024) | $0_{\pm0.0}$ | $0_{\pm0.0}$ | $0_{\pm0.0}$ | $0_{\pm0.0}$ | $0_{\pm0.0}$ | $0_{\pm0.0}$ | $0_{\pm0.0}$ |
| 3D-LOTUS (Garcia et al., 2025) | $0.3_{\pm0.3}$ | $3_{\pm4.0}$ | $0_{\pm0.0}$ | $0_{\pm0.0}$ | $0_{\pm0.0}$ | $0_{\pm0.0}$ | $0_{\pm0.0}$ |
| 3D-LOTUS++ (Garcia et al., 2025) | $17.4_{\pm0.4}$ | $76_{\pm7.4}$ | $49_{\pm8.6}$ | $37_{\pm8.1}$ | $0_{\pm0.0}$ | $0_{\pm0.0}$ | $0_{\pm0.0}$ |
| BridgeVLA (Li et al., 2025b) | $0_{\pm0.0}$ | $0_{\pm0.0}$ | $0_{\pm0.0}$ | $0_{\pm0.0}$ | $0_{\pm0.0}$ | $0_{\pm0.0}$ | $0_{\pm0.0}$ |
| CLAP | $\mathbf{31.4}_{\pm0.6}$ | $\mathbf{98}_{\pm2.7}$ | $\mathbf{87}_{\pm4.5}$ | $\mathbf{92}_{\pm5.7}$ | $0_{\pm0.0}$ | $0_{\pm0.0}$ | $0_{\pm0.0}$ |

| Method | PutItems InDrawer+4 | Tower4+1 | Tower4+3 | Stack Cups+0 | Stack Cups+3 | PutAllGroceries InCupboard+0 |
|---|---|---|---|---|---|---|
| HiveFormer (Guhur et al., 2023) | $0_{\pm0.0}$ | $0_{\pm0.0}$ | $0_{\pm0.0}$ | $0_{\pm0.0}$ | $0_{\pm0.0}$ | $0_{\pm0.0}$ |
| PolarNet (Chen et al., 2023) | $0_{\pm0.0}$ | $0_{\pm0.0}$ | $0_{\pm0.0}$ | $0_{\pm0.0}$ | $0_{\pm0.0}$ | $0_{\pm0.0}$ |
| 3D Diffuser Actor (Ke et al., 2024) | $0_{\pm0.0}$ | $0_{\pm0.0}$ | $0_{\pm0.0}$ | $0_{\pm0.0}$ | $0_{\pm0.0}$ | $0_{\pm0.0}$ |
| RVT2 (Goyal et al., 2024) | $0_{\pm0.0}$ | $0_{\pm0.0}$ | $0_{\pm0.0}$ | $0_{\pm0.0}$ | $0_{\pm0.0}$ | $0_{\pm0.0}$ |
| 3D-LOTUS (Garcia et al., 2025) | $0_{\pm0.0}$ | $0_{\pm0.0}$ | $0_{\pm0.0}$ | $0_{\pm0.0}$ | $0_{\pm0.0}$ | $0_{\pm0.0}$ |
| 3D-LOTUS++ (Garcia et al., 2025) | $0_{\pm0.0}$ | $17_{\pm10.8}$ | $30_{\pm13.4}$ | $0_{\pm0.0}$ | $0_{\pm0.0}$ | $0_{\pm0.0}$ |
| BridgeVLA (Li et al., 2025b) | $0_{\pm0.0}$ | $0_{\pm0.0}$ | $0_{\pm0.0}$ | $0_{\pm0.0}$ | $0_{\pm0.0}$ | $0_{\pm0.0}$ |
| CLAP | $0_{\pm0.0}$ | $\mathbf{30}_{\pm7.1}$ | $\mathbf{70}_{\pm6.1}$ | $0_{\pm0.0}$ | $0_{\pm0.0}$ | $0_{\pm0.0}$ |

Table 10: **Per-task Success Rate on GemBench Level 4.**

Table 11: **Ablation on number of robot demonstrations used for training.** We compare the results of training our method with different numbers of demonstrations. The results are recorded with three runs of different random seeds.

| Number of Demos | L1 | L2 | L3 | L4 | Average |
|---|---|---|---|---|---|
| 10 | $84.5 \pm 0.8$ | $81.5 \pm 0.6$ | $43.3 \pm 1.9$ | $30.5 \pm 2.1$ | $60.0 \pm 0.1$ |
| 20 | $83.9 \pm 0.3$ | $83.2 \pm 1.9$ | $49.6 \pm 2.1$ | $31.4 \pm 0.6$ | $62.0 \pm 0.5$ |
| 50 | $87.9 \pm 0.3$ | $81.2 \pm 0.1$ | $47.0 \pm 1.5$ | $28.3 \pm 2.2$ | $61.1 \pm 0.9$ |
| 100 | $86.9 \pm 1.5$ | $81.7 \pm 0.4$ | $47.8 \pm 2.6$ | $27.5 \pm 1.2$ | $61.0 \pm 0.7$ |

Table 12: **Ablation on inputs to Fine-grained Action Predictor.** We compare the results of removing some inputs to our fine-grained action predictor. The results are recorded with three runs of different random seeds.

| Inputs | L1 | L2 | L3 | L4 | Average |
|---|---|---|---|---|---|
| RGB | $83.9 \pm 1.4$ | $79.7 \pm 2.3$ | $48.2 \pm 1.3$ | $31.8 \pm 1.3$ | $60.9 \pm 0.6$ |
| RGB+Depth | $87.2 \pm 2.3$ | $83.4 \pm 1.0$ | $48.2 \pm 1.0$ | $24.3 \pm 0.9$ | $60.8 \pm 0.6$ |
| RGB+Depth+3D positional embedding (Ours) | $83.9 \pm 0.3$ | $83.2 \pm 1.9$ | $49.6 \pm 2.1$ | $31.4 \pm 0.6$ | $62.0 \pm 0.5$ |

