# OpenReview forum: "Generalizable Coarse-to-Fine Robot Manipulation via Language-Aligned 3D Keypoints"
_ICLR.cc/2026/Conference — ICLR 2026 Poster_

### Official Review · Reviewer_zNT5 · 2025-10-30

**Soundness:** 4
**Presentation:** 4
**Contribution:** 3
**Rating:** 6
**Confidence:** 3

**Summary:**

This paper proposes CLAP, a hierarchical coarse-to-fine policy for robotic 3D manipulation that integrates a pre-trained VLM as a coarse task planner and a fine-grained action predictor. The method introduces task decomposition into step-wise language instructions and employs a sequential reasoning process for 3D keypoint prediction. Evaluated on the GemBench benchmark and real-world tasks, CLAP demonstrates superior generalization to novel instructions and object variations while using significantly fewer training trajectories than prior state-of-the-art methods.

**Strengths:**

1. The approach effectively addresses compositional generalization by decomposing tasks into reusable step-wise instructions, enabling skill recombination across tasks.
2. The integration of pre-trained models with a structured reasoning pipeline enhances sample efficiency and spatial understanding, as validated through rigorous ablation studies.
3. The method achieves notable performance gains in both simulation and real-world experiments, particularly on long-horizon and articulated object tasks.

**Weaknesses:**

1. The computational overhead of the two-stage architecture and sequential VLM reasoning remains unquantified, raising concerns about real-time applicability.
2. The reliance on a chain of reasoning steps introduces fragility; errors in object localization or instruction generation may propagate and compromise task execution.
3. The method assumes tasks can be linearly decomposed into language instructions, which may not hold for tasks requiring non-sequential or implicit motor skills.

**Questions:**

1. What is the inference latency of CLAP under real-world operational conditions, and how does it compare to baseline methods like RVT2?
2. In the sequential reasoning process where the coarse task planner first localizes task-related objects before predicting step instructions and keypoints, how does the method perform when object localization fails?
3. Can CLAP handle tasks that lack clear step-wise descriptions or require fine-grained motions not easily captured by language?

---

> ### Author Response · Authors · 2025-11-21
> **Reply to Reviewer zNT5**
>
> We thank the reviewer for the clear summary of the main ideas of our paper and for acknowledging the effectiveness of our method.
> We address the reviewer's valuable comments and questions below.
> ### **W1: Computational overhead.**
> Please check the second response (R2) in meta reply.
> ### **W2: Fragility of chain of reasoning steps.**
> Yes, Chain of Thought might introduce fragility.
> However, our method allows some errors in the step-language prediction and keypoint prediction of our coarse task planner.
> Please check the fifth response (R5) in meta reply.
> One interesting future work is to incorporate RL methods to improve the robustness of the CoT reasoning.
> ### **W3: Assumption on task decomposition.**
> We acknowledge that keyframe-based imitation learning generalizes poorly to unstructured tasks.
> Actually, we can decompose different tasks in different levels.
> For structural tasks with clear steps, we can decompose them in a fine-grained level to help the agent to learn common skills.
> However, for unstructured tasks, we might use the high-level task goal without decomposition.
> Task decomposition may be a design choice we need to make when building the dataset containing different tasks.
>
> Here, we follow the prior work of using key-frame for decomposition.
> Because the tasks we consider can be decomposed and keyframe-based methods present efficiency and stability for solving these tasks.
> ### **Q1: Inference latency.**
> Please check the second response (R2) in meta reply.
> ### **Q2: Object position prediction error.**
> Considering that the prediction of the object position and the prediction of the keypoint are highly correlated, we explain the results under two situations.
> If there are some errors in the object localization but the prediction is still around the object, with noise added during training, our method can still solve the task.
> If the prediction is absolutely wrong, such as localizing the position of one object at the position of another object, then the keypoint prediction may be wrong and lead to task failure.
> ### **Q3: Decomposition of unstructured task.**
> We acknowledge that keyframe-based imitation learning may generalize poorly to unstructured tasks.
> However, as mentioned in the response to Weakness 3, we can choose how we define the task decomposition.
> The ideas in this paper are still applicable if we consider a long-horizon task (such as cleaning a room) and treat the unstructured task (such as wiping a desk) as a subtask.

---

### Official Review · Reviewer_uaCQ · 2025-10-31

**Soundness:** 3
**Presentation:** 3
**Contribution:** 2
**Rating:** 4
**Confidence:** 5

**Summary:**

This paper proposes CLAP, a hierarchical 3D manipulation policy to address the poor generalization of existing coarse-to-fine methods, especially for novel environments and skill compositions. CLAP features two innovations: 1. Coarse Task Planner: A fine-tuned VLM (Qwen 2.5 VL-3B) acts as the coarse branch. It decomposes tasks into step-wise language instructions $l_k$ and predicts a corresponding 3D keypoint $p{t_k}$ for each step, guided by a sequential reasoning process and an auxiliary object detection task. 2. Fine-grained Action Predictor: A separate module receives the specific step instruction $l_k$ and a "3D-aware representation" (built from SigLIP and DINOv2) centered at $p{t_k}$ to predict the final action. Experiments on GemBench show CLAP outperforms SOTA by 12% using only 1/5 of the data (20 demonstrations). Real-world tests also confirm strong generalization from only 10 demonstrations.

**Strengths:**

1. The VLM is innovatively fine-tuned to be the coarse branch, directly outputting 3D keypoints. The "sequential reasoning" fine-tuning pipeline is a clever method for aligning language, 3D space, and planning.
2. The method achieves strong results on GemBench (62.0% avg. success) with only 1/5 of the data (20 demos) and generalizes in the real world from only 10 demos, which is highly practical.

**Weaknesses:**

1. **Extreme architectural complexity:** This paper relies on RVT2's canonical views, uses one VLM (Qwen) as the coarse planner, and uses two other powerful pre-trained models (SigLIP and DINOv2) for the fine-grained branch. This complexity makes reproduction difficult, and it's unclear if the performance gain comes from the novel architecture or simply from stacking 3+ powerful (and separate) pre-trained encoders.
2. **Dependency on and lack of transparency about auxiliary data:** The method relies on fine-tuning on an "auxiliary 3D object detection dataset" and a "language plan dataset". The paper does not detail the source, scale, or creation cost of these datasets. If these decomposed steps require significant manual annotation, it would be a major, hidden data cost that undermines the claimed sample efficiency of 10-20 demonstrations.
3. **Dependency on keyframes:** As the authors admit in the conclusion, the entire framework is tied to keyframe-based imitation learning. This makes it unsuitable for unstructured tasks where keyframes are difficult to define (e.g., wiping, stirring), limiting its generality.
4. **Cascading errors:** This is a hierarchical system. If the coarse task planner (VLM) makes an error in the first stage—e.g., (a) generating the wrong step instruction $l_k$, or (b) predicting an inaccurate 3D keypoint $p_k$—the system will likely fail catastrophically. The paper does not discuss or evaluate the robustness to such cascading errors.
5. **Insufficient comparation with highly related work:** The "Related Work" section fails to discuss (or differentiate from) several key recent works exploring an extremely similar problem space. Specifically, this paper needs a clear comparison with GravMAD (arxiv: 2409.20154) and DeCo (arxiv: 2505.00527). All three papers (including CLAP) are trying to solve the same core challenge: how to use foundation models and task decomposition to enable imitation learning models to zero-shot generalize to novel, long-horizon, compositional 3D manipulation tasks. This lack of comparison makes it difficult for reviewers to assess CLAP's unique contribution in this increasingly crowded and important research direction.

**Questions:**

1. Please clarify the source and scale of the auxiliary data (the "language plans" and "object position dataset") required for VLM fine-tuning. Is this data auto-generated or does it require manual annotation? If manual, how does the effort (e.g., in person-hours) to create these datasets compare to collecting the 10-20 demonstrations?

2. Can you elaborate on the justification for this complex architecture? Why is it necessary to use separate SigLIP and DINOv2 encoders for the fine-grained branch, rather than (for example) re-using the VLM's (Qwen's) own visual encoder? Were simpler, encoder-sharing setups tested?

3. This framework is a cascade. How does the system perform if the coarse task planner makes an error in the first stage (e.g., predicting a keypoint $p_k$ with a 10cm offset, or generating the wrong step instruction $l_k$)? Does the fine-grained action predictor have any robustness to correct for moderate errors from the coarse planner, or does the system fail catastrophically?

---

> ### Author Response · Authors · 2025-11-21
> **Reply to reviewer uaCQ (Part 1)**
>
> We thank the reviewer for acknowledging the novelty of our designs and the effectiveness of our method.
> We address the reviewer's constructive comments and questions in detail below:
> ### **W1: Extreme architectural complexity.**
> The experimental results with Exp ID 1 in our ablation study correspond to simply training the pre-trained VLM as a task planner and include the pre-trained encoders in the fine-grained action predictor.
> The performance gain comparing this Exp ID 1 (success rate 48.0\%) with RVT2 (success rate 44.0\%) is limited.
> This experiment shows that the final performance of our method actually comes from our specific designs beyond simply stacking the pre-trained encoders.
> ### **W2: Dependency on and lack of transparency about auxiliary data.**
> We thank the reviewer for reminding us of including more details about the data.
> In the revised version, we include a paragraph in Appendix A.2 to describe the dataset in detail.
> Please check the forth response (R4) in meta reply.
> ### **W3: Dependency on keyframes.**
> We acknowledge that keyframe-based method, which predicts a next key-pose and uses a motion planner to move the gripper to it, generalizes poorly to unstructured tasks.
> However, this presents a trade-off in the choice of the action representation.
> Keyframe-based methods present efficiency and stability while predicting actions at each time step is more general and compatible for all tasks.
> ### **W4: Cascading errors.**
> This is a common issue in a hierarchical system.
> On one side, our method allows some errors in the step-language prediction and keypoint prediction of our coarse task planner.
> Please check the fifth response (R5) in meta reply.
> On the other side, it is an interesting future direction to include a mechanism for failure recovery.
> ### **W5: Insufficient comparison with highly related work.**
> We thank the reviewer for indicating the related work in 3D manipulation.
> We have updated the related work section to include the papers.
> Here, we list the key differences between our method and the other two papers.
> #### **CLAP and GravMAD**
> - CLAP is categorized as a coarse-to-fine policy where \pw{the} high-level module predicts a 3D keypoint and \pw{the} low-level module predicts the final action using the local observation located by the 3D keypoint.
> GravMAD does not fit in coarse-to-fine policies while it leverages a novel GravMap to guide the action prediction.
> - GravMAD proposes a novel task decomposition scheme to detect the sub-goal keyposes.
> CLAP follows previous work to use the heuristically defined keyposes to segment the trajectories.
> - CLAP leverages the pre-trained VLM in the task planner by finetuning it while GravMAD exploits the zero-shot ability of pre-trained modes like GPT-4o.
> Under our finetuning setting, we focus more on how to design training pipeline to effectively adapt the pre-trained VLM for keypoint prediction.
> - For simulation evaluation, GravMAD chooses a set of tasks from RLBench as the training set and creates another set of tasks to evaluate the generalization ability.
> We evaluate CLAP with GemBench, a benchmark designed for evaluating the generalization ability.
> Considering that the code of GravMAD is not released yet, we can only train and evaluate our method on the 12 base tasks used in GravMAD.
> The results are shown below:
> | Model | Average | Close Jar | Open Drawer | Meat off Grill | Slide Block | Put in Drawer | Push Buttons |
> |:---|:---:|:---:|:---:|:---:|:---:|:---:|:---:|
> | GravMAD (Manual) | 69.17 | $100.00_{\pm 0.00}$ | $76.67_{\pm 4.62}$ | $89.33_{\pm 2.31}$ | $93.33_{\pm 2.31}$ | $78.67_{\pm 6.11}$ | $98.67_{\pm 2.31}$ |
> | GravMAD (VLM) | 56.72 | $100.00_{\pm 0.00}$ | $58.67_{\pm 2.31}$ | $70.67_{\pm 2.31}$ | $80.00_{\pm 0.00}$ | $61.33_{\pm 9.24}$ | $97.33_{\pm 2.31}$ |
> | **CLAP (Ours)** | **74.9** | $100.00_{\pm 0.00}$ | $81.33_{\pm 11.55}$ | $96.00_{\pm 4.00}$ | $90.67_{\pm 6.11}$ | $93.33_{\pm 2.31}$ | $100.00_{\pm 0.00}$ |
> | | |**Stack Blocks** | **Place Cups** | **Place Wine** | **Screw Bulb** | **Insert Peg** | **Stack Cups** |
> | GravMAD (Manual) | |$56.67_{\pm 4.62}$ | $5.33_{\pm 2.31}$ | $77.33_{\pm 4.62}$ | $66.67_{\pm 6.11}$ | $32.00_{\pm 6.93}$ | $57.33_{\pm 2.31}$ |
> | GravMAD (VLM) | |$51.33_{\pm 6.11}$ | $5.33_{\pm 4.62}$ | $33.33_{\pm 4.62}$ | $54.67_{\pm 6.11}$ | $18.67_{\pm 4.62}$ | $49.33_{\pm 2.31}$ |
> | **CLAP (Ours)** | |$66.67_{\pm 2.31}$ | $6.67_{\pm 6.11}$ | $88.00_{\pm 4.00}$ | $93.33_{\pm 4.62}$ | $28.00_{\pm 6.93}$ | $54.67_{\pm 2.31}$ |
>
> Our method can achieve a higher average success rate compared to GravMAD on the 12 base tasks.

---

> > ### Author Response · Authors · 2025-11-21
> > **Reply to reviewer uaCQ (Part 2)**
> >
> > #### **CLAP and DeCo**
> > DeCo is actually built on GravMAD and improves on GravMAD.
> > So the key differences mentioned above still apply when comparing CLAP and DeCo.
> > In conclusion, Deco proposes a model-agnostic framework on extending SOTA multi-task methods like RVT2 and 3DDA.
> > In contrast, CLAP designs a novel coarse-to-fine multi-task 3D manipulation policy.
> > Considering that the code of DeCo is not released and we do not have access to the benchmark environments, we can not show a statistical comparison here.
> >
> > We thank again the reviewers for reminding the related work and cite them in the paper.
> > However, according to the last Q\&A from the ICLR 2026 reviewer guide (\url{https://iclr.cc/Conferences/2026/ReviewerGuide}) and DeCo is published on Arxiv on May 1st, 2025, we omit the detailed comparison with Deco for current version.
> >
> > ### **Q1: The source and scale of the auxiliary data.**
> > Please check the forth response (R4) in meta reply.
> > ### **Q2: Complex architecture.**
> > As mentioned in our response to Weakness 1, our ablation study shows that the final performance of our method actually comes from our specific designs beyond simply stacking the pre-trained encoders.
> > There are several reasons why we did not use the visual encoder from Qwen 2.5 VL:
> > - We explain the types of features we want to use in Section 4.2.
> > Encoders which can provide language-aligned, 3D-aware features are suitable for the task.
> > In the implementation, we instantiate our method with SigLIP and DINOv2.
> > - The visual encoder in Qwen 2.5 VL is learned from scratch using images/videos in the training set.
> > Considering that the visual inputs to our fine-grained action predictor are projected images from local point cloud, they have a large gap with the training set used to train Qwen 2.5 VL.
> > - The visual encoder in Qwen 2.5 VL has more parameters compared to the visual encoders we use.
> > A small visual encoder is enough for the fine-grained action predictor.
> > ### **Q3: This framework is a cascade.**
> > Please check the fifth response (R5) in meta reply.

---

### Official Review · Reviewer_x1Wc · 2025-11-01

**Soundness:** 2
**Presentation:** 2
**Contribution:** 1
**Rating:** 2
**Confidence:** 3

**Summary:**

This paper introduces CLAP, a hierarchical coarse to fine policy for 3D robotic manipulation that aims to generalize to novel instructions and environment changes. The coarse task planner is a fine tuned VLM that first decomposes a task into step wise language instructions, then reasons about object locations and predicts a language aligned 3D keypoint used to crop observations. The fine grained branch fuses step instruction with a 3D aware visual representation from multi view RGB D and depth encoders to predict actions. In simulation on GemBench, CLAP reports an average success rate that is 12 percent higher than the previous best method while using only one fifth of the training trajectories, and in real robot tests it generalizes to new tasks and perturbations with only ten demonstrations per task.

**Strengths:**

- CLAP augments the coarse branch into a task planner that performs step wise decomposition, object localization, and language aligned 3D keypoint prediction, then conditions the fine branch on the current step rather than the entire task. The two round inference protocol and the auxiliary 3D object detection task are thoughtful adaptations that better align VLMs to manipulation.
- Strong empirical evidence spans GemBench with detailed multi level results and an ablation table that isolates the impact of language plans, last step memory, object reasoning, and pre trained encoders. Real robot experiments further support the claims.

**Weaknesses:**

- The idea of using coarse-to-fine planning is not new, such as ReKep [1] , MakeADonut [2] and VoxPoser [3]. The technical novelty is limited and the contribution is unclear either.
- The paper lacks sufficient details about the method, e.g., how are keypoints predicted; what are their distributions; how to convert from keypoints to robot actions. The finegrained action predictor in Figure 2 is never described in detail, making it hard to understand the proposed method. Therefore, the paper seems to be unpolished and not ready to be accepted.

[1] Huang, Wenlong, et al. "Rekep: Spatio-temporal reasoning of relational keypoint constraints for robotic manipulation." arXiv preprint arXiv:2409.01652 (2024).

[2] You, Yang, et al. "Make a Donut: Hierarchical EMD-Space Planning for Zero-Shot Deformable Manipulation with Tools." IEEE Robotics and Automation Letters (2025).

[3] Huang, Wenlong, et al. "Voxposer: Composable 3d value maps for robotic manipulation with language models." arXiv preprint arXiv:2307.05973 (2023).

**Questions:**

- What fraction of your training set comes from language plans versus robot trajectories, and how were plans obtained and validated. Please report sensitivity to the amount and quality of plan data.
- How robust is the planner to plan errors or mis localized objects. Do you have a recovery strategy or re planning mechanism during execution.
- Can you provide equal trajectory comparisons to RVT2 or BridgeVLA and a study controlling for the number of demonstrations per variation.

---

> ### Author Response · Authors · 2025-11-21
> **Reply to reviewer x1Wc**
>
> We thank the reviewer for noting the key components in our method and its strong performance in both simulation and real-world scenarios.
> Below, we address the reviewer's valuable comments and questions.
> ### **W1: Novelty and contributions.**
> We take the opportunity of this remark to clarify some potential misunderstandings.
> Note that we never claim that we propose the idea of *coarse-to-fine policy*, whose definition we recall here for easier reference: ”A coarse-to-fine policy contains a coarse branch and a fine-grained branch, where the coarse branch predicts a 3D keypoint as the center to zoom in the 3D observation and the fine-grained branch uses the refined observation to predict the target action.”
> **Instead, our contributions consists in designing a novel coarse-to-fine policy method, where we finetune pretrained VLM to output language-aligned 3D keypoints to connect high-level planner and low-level action predictor**. Our design preserves the base visual scene-understanding and reasoning capabilities of pre-trained VLMs, while adapting them effectively to 3D manipulation, without requiring full end-to-end training of the VLM module as in most VLA approaches.
> We refer the reviewer to the first response (R1) in meta reply, 2nd paragraph in Introduction, 3rd paragraph in Related Work, and 3rd paragraph in the Background of original paper.
> Notably, we clearly discuss the differences between CLAP and previous coarse-to-fine methods in the Related Work.
>
> According to this definition of coarse-to-fine policy, none of the three papers mentioned by the reviewer actually lies in this category.
> Instead, these three papers propose using the strong zero-shot ability of a large pre-trained VLM (like ChatGPT-4o) to propose keypoint constraints [1], task plan [2], and 3D value map [3] respectively.
> Moreover they generate robot actions by solving optimization problems or motion planning.
>
> ### **W2: Details of the method.**
> We thank the reviewer's suggestion on providing more details about the method.
> We explain point by point below on what we already included and indicate what we revise in the updated version accordingly:
> - In the original paper, we explained how the keypoints are predicted in Section 4.1.
> Here, we further briefly introduce how the keypoints are predicted.
> Given multi-view RGB images and a task description, a pre-tranined VLM (Qwen 2.5 VL 3B) is finetuned to predict the language plan, task-related object position, next step instruction and keypoint.
> We follow  previous work for the prediction of pixel position in each view and map them to a 3D position.
> The mapping is mentioned in the last paragraph in the Background Section.
> To avoid any confusion or misunderstanding, in the revised version, we add one sentence to explicitly mention this in the first paragraph in the Method Section.
> - Following prior work, given the pixel position in each view, a Gaussian distribution is applied on the pixel position to generate a heatmap.
> - As explained in the last paragraph in the Background Section of the original paper, the keypoints predicted by our high-level module is used to zoom in the point cloud and our low-level module predicts the action given the local observation.
> Here the action is actually the next gripper pose (position, orientation and gripper open/close), and the predicted action is executed by a motion planner.
> - We list the locations where we introduce related details in the original paper.
> The basics of the fine-grained action predictor is introduced in the Background Section.
> The details of the fine-grained action predictor is introduced in the Section 4.2 of the Method Section.
> And the instantiation of this module is explained in the third paragraph in Experimental Set-up in Section 5.1.
>
> ### **Q1: Details about language plan dataset.**
> Please check the forth response (R4) in meta reply about the details of the training data.
> According to our previous experiments, our method is not sensitive to the amount of the plan data as long as the whole dataset is not dominated by the other two datasets.
> The quality of the plan data is ensured by the way we generate the plan data.
>
> ### **Q2: Robustness.**
> Our method currently does not incorporate a recovery strategy.
> It would indeed be interesting to include it as future work.
> However, our method tolerates some errors in the step-language prediction and keypoint prediction of our coarse task planner.
> Please check the fifth response (R5) in meta reply.
>
> ### **Q3: Training with different number of demonstrations.**
> Please check the third response (R3) in meta reply.
>
> [1] Wenlong Huang et al., Rekep: Spatio-temporal reasoning of relational keypoint constraints for robotic manipulation, CoRL, 2024.
>
> [2] Yang You et al., Make a donut: Hierarchical emd-space planning for zero-shot deformable manipulation with tools, RA-L, 2025.
>
> [3] Wenlong Huang et al., Voxposer: Composable 3d value maps for robotic manipulation with language models, CoRL, 2023.

---

> > ### Comment · Reviewer_x1Wc · 2025-11-25
> >
> > I appreciate the authors' efforts to clarify the novelties and contributions; the paper is now much more understandable. While some limitations remain (e.g., the dependency on keyframes), I believe this work introduces a promising manipulation method to the robotics community.

---

> ### Author Response · Authors · 2025-11-26
>
> Thank you so much for the careful consideration and raising the score! We will continue to improve the paper according to your valuable feedback.

---

### Official Review · Reviewer_EPq1 · 2025-11-01

**Soundness:** 3
**Presentation:** 3
**Contribution:** 3
**Rating:** 6
**Confidence:** 4

**Summary:**

To address the generalization issues in the hierarchical coarse-to-fine policy, the authors propose to extend a prior work, RVT2, with an VLM for task decomposition and 3D keypoint prediction. Specifically, to enhance such capabilities on the VLM, several ideas for fine-tuning the VLM have been introduced, namely 1) task decomposition, 2) VLM fine-tuning for 3D keypoint prediction, and 3) 3D-aware representation. Experimentally, the superior performance and training sample efficiency have been validated in both simulation and real-world along with comparison of a wide range of strong baselines.

**Strengths:**

- The presentation is easy-to-follow with elaborative description of the approach.

- An effective extension on top of RVT2: a sensible idea to add an VLM with enhanced generalizability of task compostionality and spatial awareness; To achieve this, the authors also propose of a set of pratical techniques, i.e., adding spatial reasoning, including previous sub-plans as inputs, 3D key-points prediction, etc..

- Superior training sample efficiency has been shown. Its performance surprisingly surpasses the SOTA methods while using only 1/5 of the training trajectories.

- Comprehensive experimental study in simulation and real-world, demonstrating significant performance gain against a wide range of baselines.

**Weaknesses:**

- Presentation seems too wordy. The writing style of the current draft reads like a technical report. I would suggest the authors to shorten the method part and make it more concise so that the readability can be further improved.

- Limited novelty. Though the performance is encouraging, the technical novelty remains limited in terms of the desgin choice and tricks applied for fine-tuning the VLM. More understanding and analysis can be conducted in order to improve the technical quality of this work, e.g., what exactly is the data efficiency coming from? Is it from the greater task compositionality or the enhanced spatial awareness?

- Lack of a explicit discussion about related works on system1&2 policies. The topic of this submission seems quite relevant to the idea of system1&2 policies, a discussion about the difference and connection would be highly recommended.

- Following the issue above, the discussion of run-time efficiency is also missing. How is it compred to the original RVT2? To which extend does adding the VLM module sacrifice the run-time efficiency?

- The fine-grained action predictor in Fig.2 is simplified, mismatched to the description in the main texts.

**Questions:**

- Run-time efficiency: have the authors tried to use the same amount of training data during training as the other baselines for further improvement?

- Redundant spatial information as input for the fine-grained action predictor: the inputs include multi-view images, depth and the 3D position embedding from pixel-wise 3D coordinates. It seems that the spatial information is quite redundant from these inputs, have the authors ablate this desgin choice?

- Generalization concern for spatial awareness of the VLM after being fine-tuned on a spefic dataset: it's a popular idea to fine-tune an VLM for better in-context generalization, e.g., task compositionality and spatial awareness. I am wondering whether the fine-tuning causes performance degradation when testing on OOD data outside the training distribution.

- Another question regarding the robustness mentioned above is about the hallucination of the VLM after being fine-tuned for greater spatial awareness. In the entropy column of table 3 [1], it is shown that spatial reasoning also helps on predictive performance on failure detection task but not necessarily on uncertainty estimation. It would be interesting and inspiring to know if this also happens when using the VLM as the task planner.

[1] Zheng, Z., Feng, Q., Li, H., Knoll, A., & Feng, J. (2024). Evaluating uncertainty-based failure detection for closed-loop llm planners. arXiv preprint arXiv:2406.00430.

---

> ### Author Response · Authors · 2025-11-21
> **Reply to reviewer EPq1**
>
> We thank the reviewer for highlighting the clarity of the presentation of our paper and the effectiveness of our proposed method.
> We have carefully considered the reviewer's constructive comments and address each point below.
> ### **W1: Presentation seems too wordy.**
> We thank the reviewer for the feedback.
> In the Method Section, we want to not only present our method but also explain the reason/intuition behind our design choices.
> We hope this can provide some insights to other researchers who work on exploiting pre-trained models in robot manipulation.
> In the camera-ready version, we will update the organization and writing to keep the key parts in the main draft and move other details to the appendix.
> ### **W2: Limited novelty.**
> About the novelty, please check the first response (R1) in meta replay.
> For the question "what exactly is the data efficiency coming from", in the paper, we focus more on explaining on the generalization ability (e.g., in the Introduction Section and in the Experiments Section).
> However, the sources of generalization and sample efficiency are similar in our method.
> Here we recall that the data efficiency is attributed to:
> 1. Task decomposition allows the agent to learn common skills among different tasks and allows compositional generalization.
> 2. Bridging VLM and 3D manipulation with projected images, separating task planning from keypoint prediction, and Chain-of-Thought reasoning lead to an effective adaptation of pre-trained VLM to the coarse task planner.
> 3. Data efficiency with respect to the robot trajectories also benefits from leveraging language plans and object position data. By co-training on these additional sources, our framework equips the model with task-planning and object-grounding abilities, leaving it to learn only how to act given sub-task instructions and known object locations. In contrast, prior approaches required the model to learn solely from robot trajectories.
>
> We include this analysis in Appendix A.5 in the revised version.
> ### **W3: Lack of related work.**
> If we understand correctly, system 1\&2 policies refer to policies which have a fast system (intuition) and a slow system (reasoning).
> Our method actually lies in this category, which includes lots of varying methods.
> Analyzing our method in this category makes it difficult to compare our method with them all and highlight the main contributions of our method.
> So we consider a subset called coarse-to-fine policies where these methods are closely related to our method.
> We introduce the related work on coarse-to-fine policies and compare our method with methods of leveraging pre-trained models to build a coarse-to-fine policy in the third part of the Related Work Section.
> ### **W4: Discussion of run-time efficiency.**
> Please check the second response (R2) in meta reply.
> ### **W5: Missing details in Figure 2.**
> We thank the reviewer so much for indicating the missing details about the fine-grained action predictor in Figure 2.
> We have updated Figure 2 in the revised version to include more details about it.
> ### **Q1: Training with the same amount of data.**
> Yes, we tried with the same number of demonstrations as the other baselines.
> Please check the third response (R3) in meta reply.
> ### **Q2: Redundant spatial information.**
> We follow the previous work of using all multi-view RGB images, depth images and 3D positions.
> Below, we show the results of the ablation study on this design choice, which are now included in Appendix A.5 in the updated version.
> From the results, we can see that removing some of the inputs lead to performance drop to some extent.
> | Inputs | L1 | L2 | L3 | L4 | Average |
> |:--|:--:|:--:|:--:|:--:|:--:|
> | RGB | $83.9 \pm 1.4$ | $79.7 \pm 2.3$ | $48.2 \pm 1.3$ | $31.8 \pm 1.3$ | $60.9 \pm 0.6$ |
> | RGB+Depth | $87.2 \pm 2.3$ | $83.4 \pm 1.0$ | $48.2 \pm 1.0$ | $24.3 \pm 0.9$ | $60.8 \pm 0.6$ |
> | RGB+Depth+3D positional embedding (Ours) | $83.9 \pm 0.3$ | $83.2 \pm 1.9$ | $49.6 \pm 2.1$ | $31.4 \pm 0.6$ | **$62.0 \pm 0.5$** |
> ### **Q3: Generalization degradation.**
> Yes, finetuning a pre-trained VLM will cause a performance drop on OOD data to some extent.
> This can also be observed in the results in the response to Q1.
> In our proposition, we use LoRA finetuning instead of fully finetuning to 1) reduce the computational cost and 2) alleviate the degradation of the generalization ability of the VLM.
> Moreover, co-training the VLM with a object position dataset which has a smaller domain shift compared to the original data used for pre-training the VLM also helps alleviate the issue.
> ### **Q4: Hallucination of VLM.**
> We do not notice similar situation as observed in this paper [1].
> However, it is interesting to incorporate a failure detector in our method to help correct the failure case.
> We leave it to future work.
>
> [1] Zhi Zheng et al., Evaluating uncertainty-based failure detection for closed-loop llm planners, ICRA workshop, 2024.

---

### Author Response · Authors · 2025-11-21
**Meta Reply (Part 1)**

We thank all the reviewers for acknowledging the strengths of our work and making thoughtful suggestions for improving the presentation.
We have revised our paper accordingly, as indicated by the blue color in the updated submission.
Moreover we have included additional details about the method and provided new experimental results in the appendix.
Below, we first address the common concerns and questions.
Then we respond to each reviewer and explain what we have revised.

### **R1: Novelty and contributions.**
While coarse-to-fine policies have demonstrated strong sample efficiency in 3D manipulation, their generalization remains limited, even when combined with pre-trained models.
We introduce a new coarse-to-fine architecture that differs from prior coarse-to-fine methods and existing VLAs: we finetune pretrained VLM to output language-aligned 3D keypoints to connect high-level planner and low-level action predictor.
Our design preserves the base visual scene-understanding and reasoning capabilities of pre-trained VLMs, while adapting them effectively to 3D manipulation, without requiring full end-to-end training of the VLM module as in most VLA approaches.
Our approach achieves both high sample efficiency and strong generalization, validated through extensive simulation and real-world experiments.

### **R2: Run-time Efficiency.**
Indeed including the VLM module increases computation costs, however the effective impact on control time may be limited (see explanation below).
Note that longer computational time is a common issue when scaling up a model.
Here, we provide additional statistics about the model size, inference time, and real-world episode time.
This is tested on a RTX 4090 GPU and we have not optimized the inference of the VLM.
| | Coarse Branch | | | Fine-grained Branch | |  | Whole System|
|:---|:---:|:---:|:---:|:---:|:---:|:---:|:---:|
| **Model** | **All Parameters** | **Trainable Parameters** | **Inference Time** | **All Parameters** | **Trainable Parameters** | **Inference Time** | **Episode Time** |
| RVT2 | 36M | 36M | 10ms | 37M | 37M | 20ms | **~18s** |
| CLAP | 3775M | 20M | 3s | 1000M | 36M | 40ms | **~20s** |

Note that, we apply an asynchronous inference mode [1] during real-world experiments to avoid the delay of a large policy.
Intuitively, the policy predicts multiple future actions instead of only one action.
The action sequence is stored in a buffer and the robot will execute the predicted actions one by one.
Given the new observation after executing one action, the policy will predict new actions to update the action buffer.
Under this setting, as long as the execution time of an action (moving the gripper to next key pose using motion planner) is longer than the inference time of the model, which is the common case under our real-world experiments, there will not be a significant delay.
Moreover, although we incorporate large pre-trained models, the training time decreases compared to the baseline model like RVT2, due to the usage of LoRA finetuning and faster convergence with fewer robot trajectories.

### **R3: Study on number of demonstrations for training.**
Here are the results of training our method with different numbers of demonstrations.
We can find that further increasing the number of robot trajectory improves on the in-domain performance (L1) while this does not help in the average success rate.
The main reason is that the diversity of the trajectories in the training data is limited (only variation on the object positions).
Therefore, the generalization ability of our method does not solely rely on the robot trajectory.
Overall, our method is sample efficient and robust to different number of demonstrations.
We also include the results and analysis in Appendix A.5 in the updated version.
| Number of Demos | L1 | L2 | L3 | L4 | Average |
|:---:|:---:|:---:|:---:|:---:|:---:|
| 10 | $84.5 \pm 0.8$ | $81.5 \pm 0.6$ | $43.3 \pm 1.9$ | $30.5 \pm 2.1$ | $60.0 \pm 0.1$ |
| 20 | $83.9 \pm 0.3$ | $83.2 \pm 1.9$ | $49.6 \pm 2.1$ | $31.4 \pm 0.6$ | **$62.0 \pm 0.5$** |
| 50 | $87.9 \pm 0.3$ | $81.2 \pm 0.1$ | $47.0 \pm 1.5$ | $28.3 \pm 2.2$ | $61.1 \pm 0.9$ |
| 100 | $86.9 \pm 1.5$ | $81.7 \pm 0.4$ | $47.8 \pm 2.6$ | $27.5 \pm 1.2$ | $61.0 \pm 0.7$ |

[1] Mustafa Shukor et al., Smolvla: A vision-language-action model for affordable and efficient robotics, URL https://arxiv.org/abs/2506.01844, 2025.

---

> ### Author Response · Authors · 2025-11-21
> **Meta Reply (Part 2)**
>
> ### **R4: Additional details about training data.**
> We have three datasets in total for finetuning the pre-trained VLM: a language plan dataset, an object position dataset and a robot trajectory dataset.
> - For the language plan dataset, we label each task with a language plan according to the decomposition of the task.
> We augment the data by asking an LLM to provide some variations on the language descriptions of the plan.
> This is acceptable for limited number of tasks. If we want to scale up the method, one solution would be to apply a pre-trained VLM to label the trajectory.
> - For the object position dataset, we randomly initialize the scene of the tasks and apply transformations (translation and rotation) on the scene.
> We record the RGB-D images along with the 3D positions of the objects.
> The 3D positions of the objects are projected into canonical views to obtain the pixel positions in each view.
> In real-world experiments, there are several options to collect this dataset: 1) Use the end-effector of the robot arm to record the 3D position of the objects; 2) Use pre-trained segmentation model like SAM2 [2] or object detection model like Grounding-DINO [3] to label the object in each view; 3) Use open-source object position dataset like RoboPoint [4].
> None of these options introduces significant additional efforts for data collection.
> - For the robot trajectory dataset, we explain in Section 5.1 on how we sample transitions from the demonstrations to build observation-action pair.
> Transformations (translation and rotation) are also applied on the transitions to augment the data.
>
> In practice, the number of training samples for each dataset used for GemBench is listed below:
> | | Language Plan | Object Position | Robot Trajectory Data |
> |:---|:---:|:---:|:---:|
> | **Train** | 30515 | 39777 | 169665 |
> | **Validation** | 6103 | 16650 | 17400 |
>
> In the revised version, we include a paragraph in Appendix A.2 to describe the datasets in detail.
>
> ### **R5: Robustness.**
> Our method tolerates some errors in the step-language prediction and keypoint prediction of our coarse task planner.
> 1. Data augmentation is applied on the step-language instructions.
>     And the low-level instruction is processed by a frozen pre-trained encoder.
> 2. Noise is added to the ground-truth keypoints when training the fine-grained action predictor.
>
> [2] Nikhila Ravi et al., Sam 2: Segment anything in images and videos, URL https://arxiv.org/abs/2408.00714, 2024.
>
> [3] Shilong Liu et al., Grounding dino: Marrying dino with grounded pre-training for open-set object detection, ECCV, 2024.
>
> [4] Wentao Yuan et al., Robopoint: A vision-language model for spatial affordance prediction in robotics, CoRL, 2024.

---

### Meta-Review · Area_Chair_xkwK · 2026-01-08

**Summary:**

This work presents CLAP, a method for object manipulation. It uses a coarse-to-fine scheme based on RVT-2, an LLM for task decomposition, and a fine-tuned VLM for keypoint prediction. It demonstrates superior generalization performance, achieving SOTA on GemBench and better generalization in the real world. The reviewers were initially borderline mixed, but the AC finds that the authors did a good job in addressing all the major concerns raised. The AC does not see any major remaining concerns and recommends accepting the paper. The authors should revise the paper accordingly, taking into account the reviews.

**Reviewer Concerns:**

Addressed:

- Technical Novelty: The AC feels the authors justified their contributions well in the rebuttal, clarifying that the main contribution is not proposing coarse-to-fine planning itself, but rather a 3D manipulation framework that generalizes better. Concerns regarding ReKep and VoxPoser do not seem to hold much ground.
- Run-Time Efficiency: Addressed well.
- Extreme Architectural Complexity: Addressed via new ablations.
- Comparison with other related works: Addressed.

Limitations such as dependence on keyframes and the hierarchical nature of the system are acknowledged.

**Reviewer Scores:**

Reviewer x1Wc -> from 2 to 4 or 6
Reviewer EPq1 -> 6 or higher
Reviewer uaCQ -> 4 to 6
Reviewer zNT5 -> likey 6

---

### Decision · Program_Chairs · 2026-01-26

Accept (Poster)